# Neutrophil polarization by IL-27 as a therapeutic target for intracerebral hemorrhage

Xiurong Zhao[1], Shun-Ming Ting[1], Chin-Hsuan Liu[1], Guanghua Sun[1], Marian Kruzel[2], Meaghan Roy-O'Reilly[1] & Jaroslaw Aronowski[1]

Shortly after intracerebral hemorrhage, neutrophils infiltrate the intracerebral hemorrhage-injured brain. Once within the brain, neutrophils degranulate, releasing destructive molecules that may exacerbate brain damage. However, neutrophils also release beneficial molecules, including iron-scavenging lactoferrin that may limit hematoma/iron-mediated brain injury after intracerebral hemorrhage. Here, we show that the immunoregulatory cytokine interleukin-27 is upregulated centrally and peripherally after intracerebral hemorrhage. Data from rodent models indicate that interleukin-27 modifies neutrophil maturation in the bone marrow, suppressing their production of pro-inflammatory/cytotoxic products while increasing their production of beneficial iron-scavenging molecules, including lactoferrin. Finally, interleukin-27 or lactoferrin administration results in reduced edema, enhanced hematoma clearance, and improved neurological outcomes in an animal model of intracerebral hemorrhage. These results suggest that interleukin-27/lactoferrin-mediated modulations of neutrophil function may represent a therapeutically viable concept for the modification of neutrophils toward a "beneficial" phenotype for the treatment of intracerebral hemorrhage.

[1] Department of Neurology, University of Texas Health Science Center at Houston, McGovern Medical School, Houston, TX 77030, USA. [2] Department of Integrative Biology and Pharmacology, University of Texas Health Science Center at Houston, McGovern Medical School, Houston, TX 77030, USA. Correspondence and requests for materials should be addressed to J.A. (email: J.Aronowski@uth.tmc.edu)

ntracerebral hemorrhage (ICH) is a devastating form of stroke with 30–67% mortality and poor prognosis for which no effective therapy is currently available[1–3]. Rapid accumulation of blood within the brain parenchyma causes compression and primary "mechanical" brain damage. Only half of ICH-related deaths occur within the first 2 days[4], stressing the importance of secondary brain injury in ICH pathogenesis, including the toxicity of hemolytic products (e.g., hemoglobin/heme/iron), oxidative stress, and inflammation[1, 2, 5–9]. After ICH, microglia are activated within minutes[2] to release cytokines, chemokines, and proteases, which coordinate the recruitment of potentially damaging polymorphonuclear neutrophils (PMN) from the periphery into the ICH-affected brain[10, 11].

While the first wave of PMNs enter the brain within hours, PMNs continue to enter the ICH-affected site for several days after ICH onset[11]. Once inside the brain, PMNs release various microbial defense-related molecules that could aggravate ICH pathogenesis, consistent with the finding that depletion of PMNs prior to ICH could mitigate ICH-mediated injury[12]. However, PMNs also release a subset of molecules that might benefit the ICH-compromised brain, including iron-sequestering lactoferrin (LTF) and hemoglobin-sequestering haptoglobin (Hp)[13, 14]. This ratio of damaging to beneficial molecules may be important in ICH pathogenesis and treatment. However, the relevance of beneficial PMN-derived molecules in ICH is not completely clear and is, in part, the subject of this report.

PMN maturation takes place in bone marrow (BM), where developing PMNs (BM-PMN) accomplish the majority of protein transcription, synthesis, and packaging into intracellular granules[15]. This maturation process defines the properties (chemical composition, including LTF content) of mature PMNs. After reaching terminal differentiation, PMNs are released from the BM into circulation. Upon exiting the marrow, they retain their BM-established phenotype and continue with only limited and specific renewal of key proteins. Transcripts for many proteins, including LTF, are not detected in mature PMNs[16–19]. After injury, PMNs carry their prepared granule content to the site of inflammation (here, the ICH-affected brain).

Interleukin-27 (IL-27) is a heterodimeric cytokine composed of IL-27 p28, which is unique to IL-27, and Epstein-Barr Virus-Induced Gene 3 Protein (EBI3), a structural component of both IL-27 and IL-35[20, 21]. IL-27 acts on various cell types, including T cells, B cells, and macrophages via a heterodimeric IL-27 receptor (IL-27R) composed of IL-27Rα and gp130[22]. IL-27 has many activities, including the unique ability to limit inflammation[23] and immune-mediated pathology associated with autoimmune responses[24, 25]. By contrast, elevated expression of IL-27 is reported in some pathological autoimmune conditions, including rheumatoid arthritis[26], psoriasis[27], and multiple sclerosis[28]. IL-27 is mainly produced by activated dendritic cells and macrophages, but has also been reported to be produced by astrocytes[25, 29]. Interestingly, the IL-27R is also present on PMNs[20, 30, 31]. Although there is very limited data, IL-27 and IL-27R on PMNs has been proposed to contribute to the negative regulation of reactive oxygen species and cytotoxic granule component production[30, 31]. IL-27 also modulates haematopoietic stem cells' differentiation into myeloid progenitors[22, 32], suggesting a regulatory function of IL-27 in modulating PMN maturation and phenotype.

Therefore, we hypothesize that ICH induces the production of immunoregulatory IL-27. Indeed, IL-27 signals to maturing PMN in the BM, enhancing their production of potentially beneficial proteins (including iron-sequestering LTF) and induces a less harmful PMN phenotype that may benefit the ICH-compromised brain. We thus demonstrate that pharmacological approaches adopted from this pathway (IL-27 and LTF supplementation) could be considered as potential therapeutic strategies to mitigate ICH-mediated damage and to improve post-ICH recovery.

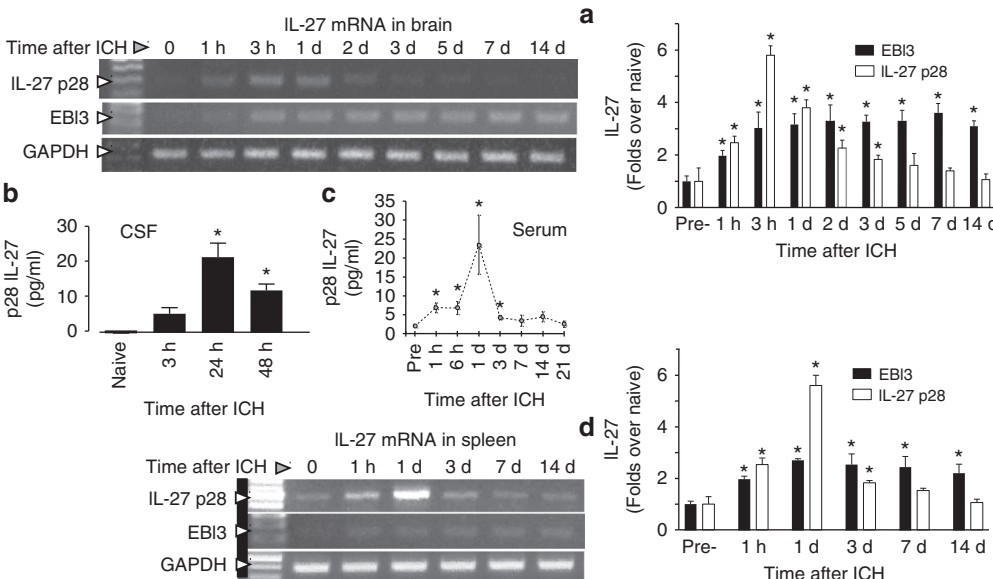

**Fig. 1** Increased IL-27 production after ICH. **a** Photograph of representative gels demonstrating temporal changes of IL-27 p28 and EBI3 mRNA in the ICH-affected corpus striatum, at 0 h (sham control) and 1 h to 14 days after ICH in SD rats, which was assessed using RT-PCR, and (**a**, *right panel*) a bar graph of densitometrical quantitation. GAPDH was used as an internal control. The data were calculated as mean ± SEM ($n = 4$). **b** Bar graph of IL-27 p28 protein levels in CSF in naïve rats and in rats at 3, 24, and 48 h after ICH. The data were calculated as mean ± SEM ($n = 3$/time point). **c** Bar graph showing timecourse of IL-27 p28 protein levels in blood serum at 1 h to 21 days after ICH in C57BJ6 mice. The data were expressed as mean ± SEM ($n = 5$). **d** Photograph of a representative gel demonstrating temporal changes in IL-27 p28 and EBI3 mRNA in the spleen of rats subjected to ICH, assessed using RT-PCR, and (**d**, *right panel*) a bar graph of densitometrical quantitation. GAPDH was used as an internal control. The data were calculated as mean ± SEM ($n = 4$). *$p \leq 0.05$, compared with the Sham group (0 h) for all the panels. The *p*-value in this figure was generated using one-way ANOVA followed by Newman–Keuls post-test

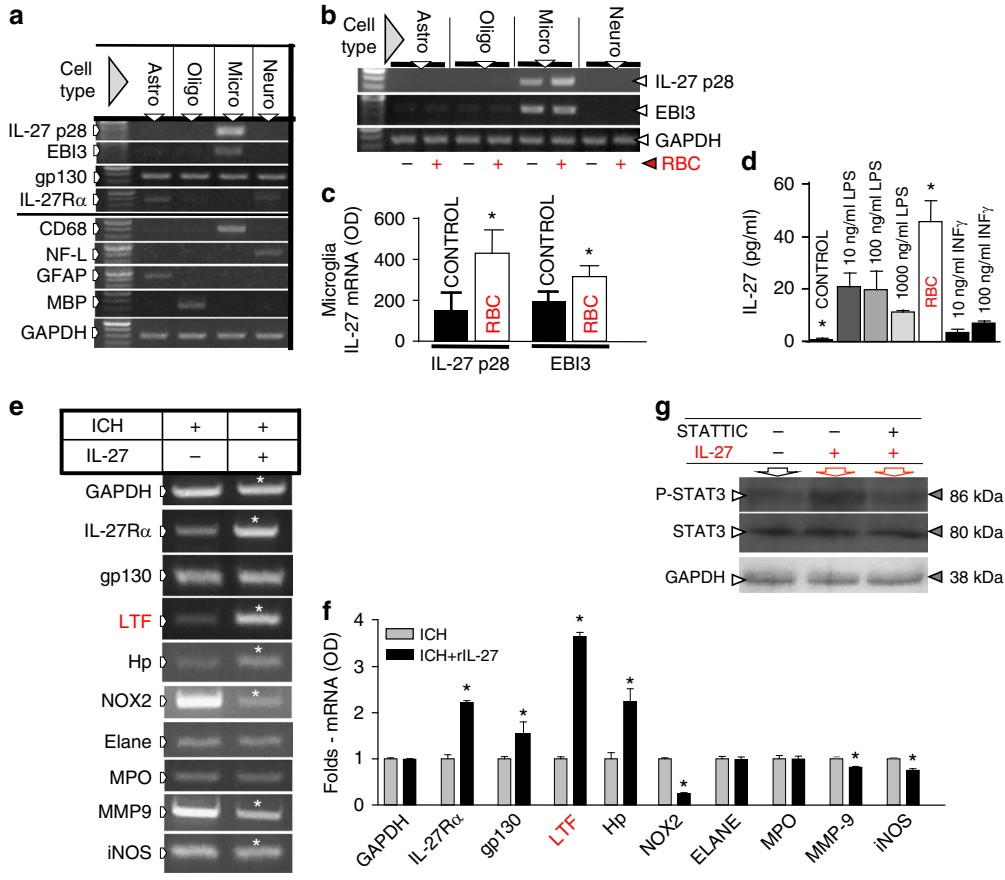

**Fig. 2** Microglia-derived IL-27 causes neutrophil polarization. Cell-specificity of IL-27 and IL-27R expression in the SD rat brain cells (**a–d**), and the gene expression profile (**e**, **f**) in IL-27-modified mouse BM-PMN precursors, and **g** STAT3/pSTAT3 protein in rat BM-PMN exposed to rIL-27, in vitro. **a** Photograph of representative gels demonstrating IL-27 p28 and EBI3 mRNA and IL-27Rα mRNA and gp130 mRNA expression in primary rat brain microglia (micro, identified with CD68), neurons (neuro, identified with neurofilament L, NF-L), astrocytes (astro, identified with glial fibrillary acidic protein, GFAP), and oligodendrocytes (oligo, identified with myelin basic protein, MBP) in culture. IL-27 p28 and EBI3 are exclusively expressed by microglia. The gp130 is expressed by all cell types, while IL-27Rα is mainly by astrocytes and neurons. GAPDH was used as an internal control. **b**, **c** Representative RT-PCR gels and bar graph of IL-27 p28 and EBI3 mRNA expression in rat brain cells in response to RBC. IL-27 p28 and EBI3 are upregulated in rat microglia (Micro), but not in astrocytes (Astro), oligodendrocytes (Oligo) or neurons (Neuro) at 6 h after exposure to RBC. The data were calculated as mean ± SEM, $n = 4$–6. *$p \leq 0.05$ (using paired $t$-test), compared with the media control group (without RBC). GAPDH was used as an internal control. **d** IL-27 p28 protein determined with ELISA in rat microglial culture medium at 6 h after exposure to RBC, LPS, and interferon gamma (IFNγ). *$p \leq 0.05$, compared with the control group ($n = 3$), established using one-way ANOVA followed by Newman–Keuls post-test. **e** Photograph of RT-PCR gels and **f** bar graph demonstrating the gene expression profile in the PMNs harvested from mouse BM-PMNs at 24 h after ICH and then cultured in 10% mouse serum in RPMI1640 and treated with recombinant mouse IL-27 (150 pg/ml) or saline for 24 h, in vitro. The analyzed genes included: iNOS, MMP-9, NOX2, Hp, LTF, myeloperoxidase (MPO), neutrophil elastase (ELANE), and GAPDH. Data were expressed as mean ± SEM ($n = 3$–5). *$p \leq 0.05$, compared with the vehicle-treated group, established using paired $t$-test. **g** Photos of immune Western blotting of phosphorylated form (pSTAT3) and total STAT3 protein in purified rat BM-PMN in culture at 6 h after incubating in recombinant mouse IL-27 (150 pg/ml) with or without 10 μM STATTIC. GAPDH was used as an internal control

## Results

**ICH induces production of IL-27 in the brain and periphery**. IL-27 is an immunoregulatory cytokine that may modulate ICH-induced immune responses. Using a clinically relevant animal model, we induced ICH and measured the expression of IL-27 over time. mRNA analysis showed that IL-27 p28 in the brain increased within 1 h of ICH onset, peaked around 3 h, and then declined over the next 2 days. The EBI3 increased similarly to p28; however, its induction was long lasting and persisted for at least for 14 days (Fig. 1a). The transient nature of IL-27 p28 gene induction in the brain is mirrored by the transient elevation (peaking at 1d) of IL-27 p28 protein in the CSF (Fig. 1b) and in peripheral blood (Fig. 1c). Unexpectedly, ICH-induced IL-27 gene expression is elevated in the remotely positioned spleen (Fig. 1d), with a temporal expression profile similar to that seen in the brain, suggesting IL-27 protein elevations detected in

peripheral blood could originate from multiple sources, in both the brain and periphery.

**Microglia are the source of brain-derived IL-27**. Prompted by the animal studies showing a robust increase in IL-27 production in the ICH-affected brain, we used cultured astrocytes, oligodendroglia, microglia, and neurons (with and without exogenously added red blood cells (RBCs) to mimic ICH in vitro) to establish that microglia are the dominant cell type expressing IL-27 mRNA (Fig. 2a–c). We established that the addition of RBCs to microglia (a process that normally leads to engulfment of RBC by microglia[6], a critical step in post-ICH brain repair) augments the mRNA synthesis of both IL-27 subsets only in microglia, but not in other brain cell types (Fig. 2b, c). Finally, adding RBCs to microglial culture augments IL-27 production and release, as

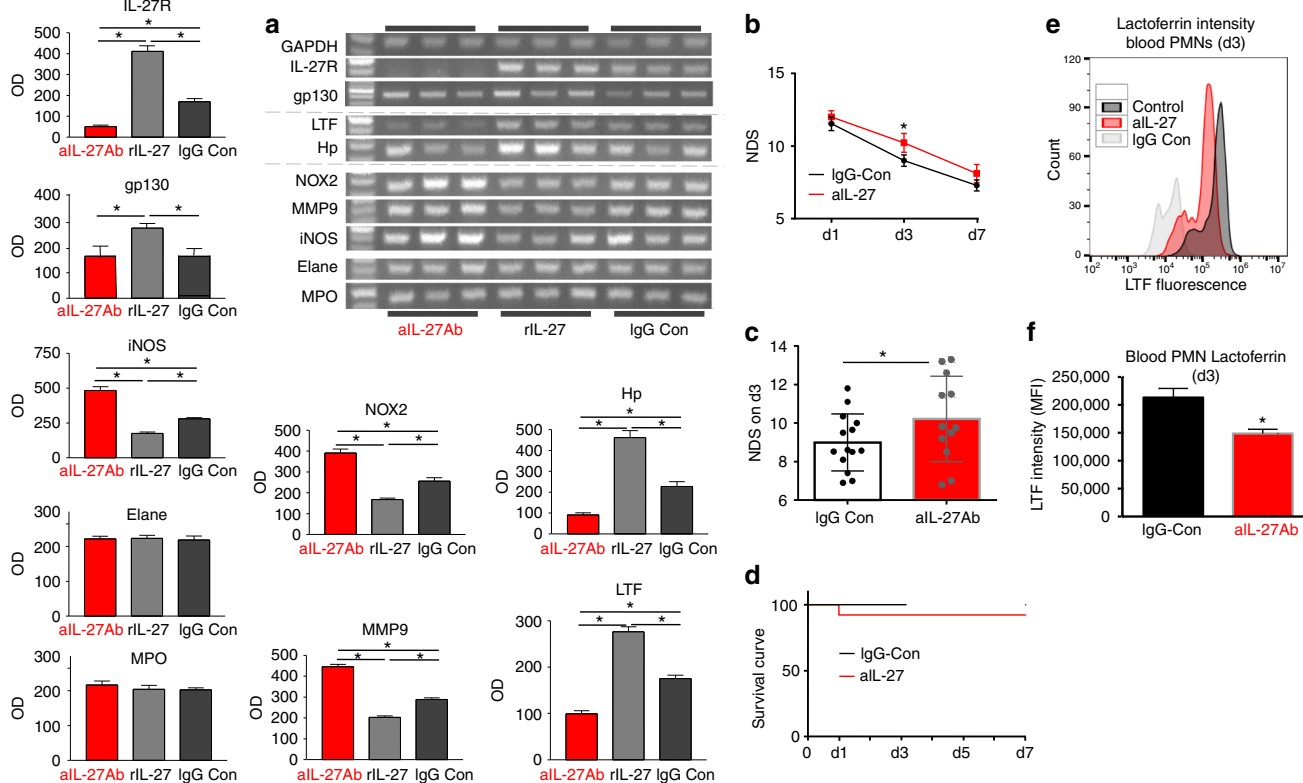

**Fig. 3** IL-27 neutralization in C57BJ6 mice alters gene expression in BM-PMNs and aggravates neurological outcome after ICH. **a** Photograph of representative gels (RT-PCR) and bar graphs demonstrating the expression profile of indicated genes in the BM-PMNs, purified from mice at 48 h after ICH that were treated with either mouse anti-IL-27 antibody (αIL-27Ab, 40 µg/mouse), recombinant mouse IL-27 (rIL-27; 50 ng/kg), or rat IgG isotype control antibody (40 µg/mouse), all by i.v. injection at 0.5, 24, and 42 h after ICH. Data are expressed as mean ± SEM ($n = 5$). *$p \leq 0.05$, established using one-way ANOVA followed by Newman–Keuls post-test. **b** The NDS in mice treated with anti-IL-27 antibody (αIL-27Ab) and rat IgG isotype control (IgG Cont) at days 1, 3, and 7 after ICH. The grand NDS is a composite score from four behavioral tests (Footfault, Postural Flexing, Wire, and Circling). *$p < 0.05$, established using one-way ANOVA followed by Newman–Keuls post-test. **c** Bar graph of the NDS on day 3 from the above study. In **b**, **c**, the data were expressed as mean ± SEM ($n = 12$). *$p < 0.05$, compared with the control, established using paired $t$-test. **d** Survival curve of day 1–day 7 after ICH in the above study in **b**. There is no difference between the groups. **e**, **f** Flow cytometry of peripheral blood taken from animals treated with anti-IL-27 antibody (αIL-27) or isotype control IgG (IgG Con) at three days post-ICH and stained for LTF. Data presented as MFI and given as mean +/− SEM ($n = 3$/group), established using paired $t$-test. An isotype control antibody (IgG Con) was used to determine the negative and positive gates for flow cytometry

determined by measuring IL-27 content in the culture media (Fig. 2d). By comparing RBCs to lipopolysaccharide (LPS) or INFγ, two known inducers of IL-27[33], we showed that RBCs are uniquely effective in inducing IL-27 production in microglia (Fig. 2d). This demonstrates that RBC-activated microglia produce and secrete IL-27, likely contributing to the overall increase of IL-27 mRNA in the brain and IL-27 protein seen in the CSF in animals after ICH (Fig. 1a).

**IL-27 modifies PMN phenotypes in BM**. IL-27 is a pleiotropic cytokine that acts through IL-27R on PMNs[30], cells that play a central function in ICH pathogenesis[11]. We next examined whether IL-27 could modify the phenotype of developing BM-PMNs. BM-PMNs have wheel-like nuclei and constitute 5–20% of the total nucleated cells in BM. We purified these BM-PMNs from mice 24 h after ICH and immediately cultured the cells in vitro, incubated with or without recombinant IL-27 (rIL-27). After 24 h, cells were analyzed for gene expression of pre-selected PMN components that could affect ICH pathogenesis. We detected downregulation of pro-inflammatory genes, inducible nitric oxide synthase (iNOS), matrix metallopeptidase-9 (MMP-9), and NADPH oxidase 2 (NOX2), and upregulation of anti-inflammatory genes, hemoglobin-neutralizing Hp and iron-

sequestering LTF, with rIL-27 (Fig. 2e, f). Interestingly, IL-27 robustly increased the expression of its own receptor, IL-27R (both of alpha subunit and gp130), suggesting that IL-27 can amplify its own signal transduction in PMNs. Exposure to rIL-27 did not affect production of MPO, ELANE, or glyceraldehyde-3-phosphate dehydrogenase (GAPDH), indicating that the effect of IL-27 was target-specific (Fig. 2e, f). In agreement with the utilization of canonical signaling pathway by IL-27 in BM-PMNs, the exposure to IL-27 caused increased phosphorylation of STAT3 that was effectively reversed by a STAT3 inhibitor, STATTIC (Fig. 2g).

We next asked whether the gain or loss of IL-27 after ICH in vivo could augment or reverse this BM-PMN phenotype. Thus, we injected mice after ICH with rIL-27, anti-IL-27 antibody (to neutralize circulating IL-27 p28), or isotype IgG control and 2 days later harvested BM-PMN for gene analysis. IL-27 neutralization effectively reduced the amount of native IL-27 in the blood plasma by 81% ($p < 0.0003$; $n = 6$). As anticipated, BM-PMNs isolated from mice treated with anti-IL-27Ab, as compared to BM-PMNs harvested from control mice, showed increased pro-inflammatory gene expression and reduced expression of Hp, LTF, and IL-27R (Fig. 3a). rIL-27 showed the opposite effect. Again, no effect on myeloperoxidase, ELANE, or GAPDH expression was noted. Ultimately, anti-IL-27Ab-treated mice

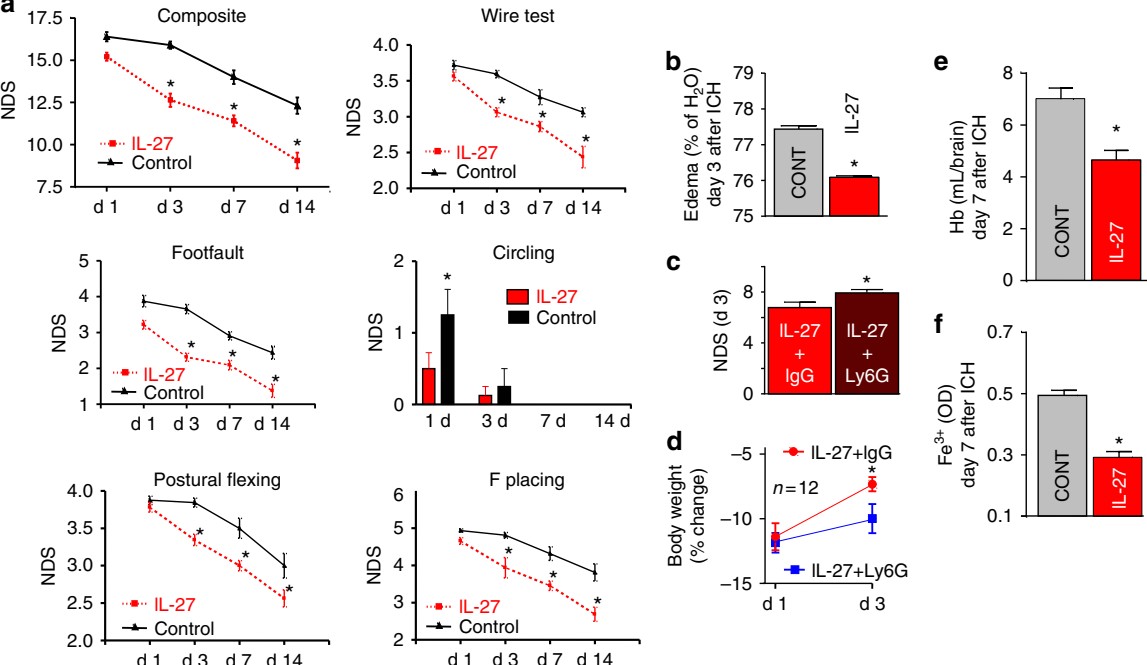

**Fig. 4** The therapeutic effect of rIL-27 in a clinically relevant mouse model of ICH. ICH in mice was induced using autologous blood injection. The rIL-27 treatment was initiated at 30 min after ICH by i.v. injection and then daily by s.c. injection for 6 days, each treatment at 50 ng/kg. The outcome include: **a** the individual behavioral/NDS of footfault, circling, wire, postural flexing, and forward placing test, and grand NDS, a composite NDS score on day 1–14 after ICH ($n = 8$) The data were expressed as mean ± SEM and *$p < 0.05$ was established using one-way ANOVA followed by Newman–Keuls post-test. **b** Brain edema (water content) on day 3 ($n = 5$); **e** hemoglobin (Hb) and **f** iron content in ICH-affected brain (index of hematoma removal) on day 7 ($n = 5$). **c** The grand NDS on day 3 after ICH in mice treated with rIL-27 and with Ly-6G or rat isotype IgG, control. Twenty-four of C57/BJ6 mice were treated with rIL-27 (Biolegend, 577404, a heterodimer of mouse recombinant IL-27 EBI3 and p28 linked by a GGGSGGGSGGGTGGGS linker), 50 ng/kg by i.v. at 30 min and then daily s.c., on day 1–3 after ICH. To deplete neutrophils, 12 mice were injected with Ly-6G, a neutrophil neutralizing antibody (BioXCell, 1A8, BE0075) at 500 µg/mouse at 2, 24, and 72 h after ICH (i.p.). And 12 mice were injected with rat isotype IgG (IgG2a, BioXCell, 2A3, BE0089). The NDS were quantified with three behavioral tests (postural flexing, circling, and footfault) 3 days after ICH. **d** The percent change of body weight on day 3 (compared with the body weight on day 0, right prior to ICH). In **b**–**f**, all data were expressed as mean ± SEM. *$p \leq 0.05$, compared with the corresponding vehicle control, established using paired $t$-test

showed worse neurological deficits score (NDS) as compared to animals treated with control antibodies (Fig. 3b, c), suggesting beneficial functions of IL-27 in ICH. Anti-IL-27Ab-treatment did not significantly affect animal mortality after ICH (Fig. 3d). In addition, flow cytometry on blood taken from animals at day 3 after stroke demonstrated that anti-IL-27-Ab significantly decreased the intensity of LTF per blood neutrophil ($p = 0.02$; $n = 3$/group) after ICH (Fig. 3e, f).

**rIL-27 treatment improves ICH outcomes.** To test whether IL-27 administration improves outcome after ICH, we subjected mice to ICH and then treated with rIL-27. rIL-27 reduced ICH-induced NDS, as demonstrated by several behavioral tests (Fig. 4a), and also reduced brain edema (Fig. 4b). There was no effect of IL-27 treatment on the number of PMNs infiltrating the ICH-affected brain ($12.2 \pm 4.5$ in control vs. $12.0 \pm 4.1$ in rIL-27-treated), as determined on day 2. To verify whether this IL-27-mediated benefit requires the presence of PMNs, we depleted PMNs (44% reduction) by the administration of an anti-Ly6G antibody in the IL-27-treated animals. We found that the reduction of PMNs was associated with a modest but significant loss of rIL-27 efficacy in reducing NDS (Fig. 4c) and the compromised body weight recovery (Fig. 4d), suggesting that PMNs are, in part, required for the protective effects of IL-27 treatment after ICH. The most intriguing by-product of rIL-27 treatment was the improved removal of hemoglobin (Fig. 4e) and free iron (Fig. 4f) from the ICH-affected brain. This suggests that sequestration of pro-oxidative hemoglobin and iron may represent an

underlying mechanism by which IL-27 improves outcome. To explore how IL-27 mediates improved iron clearance, we focused on the iron-binding LTF, a gene product that was upregulated in BM-PMNs with IL-27 treatment (Fig. 2f).

**rIL-27 helps the expression of iron-neutralizing LTF in PMNs.** Since PMNs are the main physiological source of LTF (a protein with an exceptionally high affinity for sequestering iron), we hypothesized that IL-27-modified PMNs could enhance the delivery of LTF to the ICH-affected brain, augmenting the neutralization of toxic iron and improve ICH outcome.

First, using immunohistochemistry (Fig. 5a) or FACS staining (Fig. 5b, c), we confirmed that LTF is indeed present in BM-PMNs. Next, we demonstrated that treating animals or freshly isolated BM-PMNs with rIL-27 upregulates LTF expression by BM-PMNs (Figs. 2f, 3a). We also harvested BM cells after ICH from mice treated with rIL-27 or saline and analyzed the cells using flow cytometry. We established that IL-27 treatment does not alter the proportions of inflammatory monocytes (3.9 vs. 4.45%), T cells (2.6 vs. 2.0%), or PMNs (59 vs. 61%) present in the BM. We also confirmed that only PMNs (CD45$^+$CD11b$^+$Ly6C$^+$Ly6G$^+$ cells) produce LTF (Fig. 5b). Finally, we found that the levels of LTF per PMN (as measured median fluorescence intensity, MFI) is significantly higher in PMNs from animals treated with rIL-27 (Fig. 5b, c), thus providing additional evidence that IL-27 treatment could increase LTF content of BM-PMNs in animals after ICH.

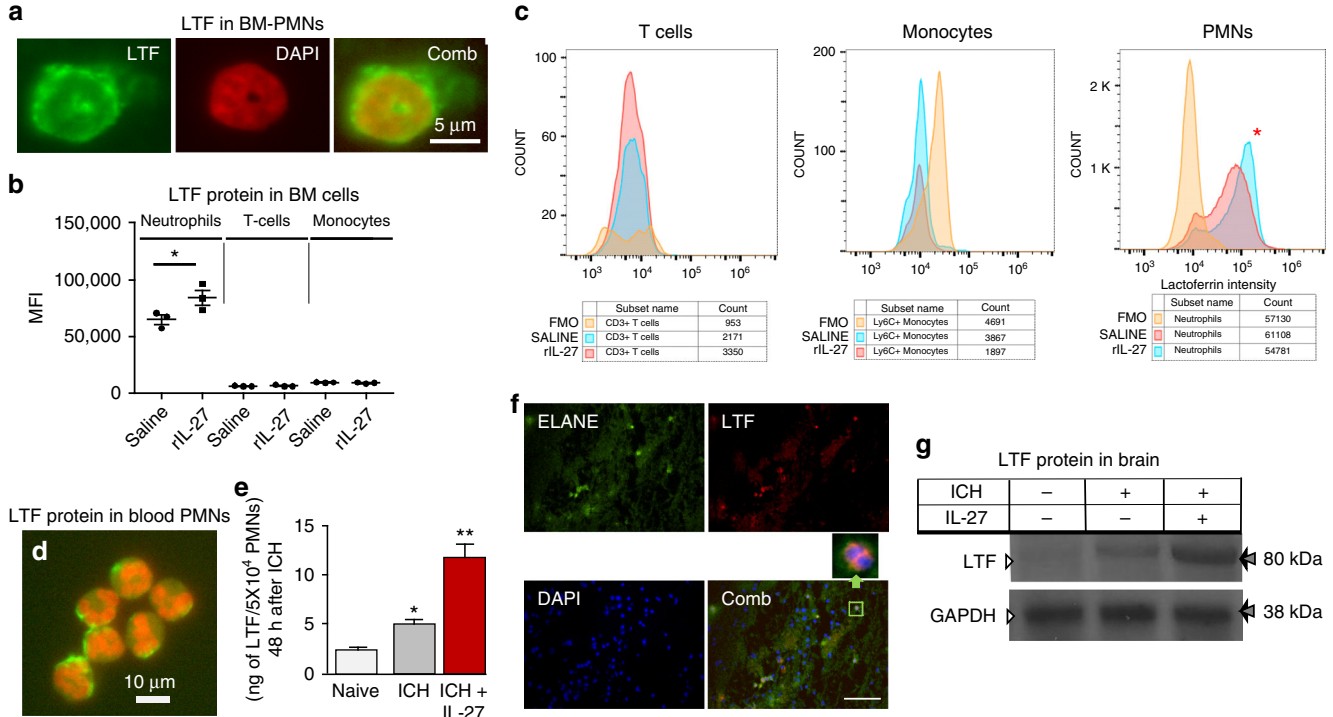

**Fig. 5** IL-27 induces LTF in PMNs. **a** Photomicrograph of representative LTF, immunostaining (*green*) in normal mouse BM-PMNs. The nuclei are stained with DAPI (*red*). **b**, **c** MFI of LTF, staining per cell was determined in BM-PMNs by flow cytometry, with T cells and monocytes serving as negative controls. Analyzed cells were from mice treated with saline or rIL-27 (50 ng/kg, i.v., at 30 min, 24 h, 42 h after ICH) and harvested at 48 h after the ICH. **d** Photomicrograph of representative LTF staining (*green*) in peripheral blood mature PMNs. The nuclei were stained with DAPI. *Scale bar* = 10 μM. **e** Bar graph of LTF, protein content (established with ELISA) in peripheral blood PMNs in naïve rats, and rats 48 h after ICH with and without treatment with rIL-27 (50 ng/kg, i.v., at 30 min, 24 h, 42 h after ICH). Data are expressed as mean ± SEM (*n* = 5). \**p* ≤ 0.05, vs. naïve rat, \*\**p* ≤ 0.05, vs. naïve rats and rats after ICH without IL-27 treatment. The *p*-value was established using one-way ANOVA followed by Newman–Keuls post-test. **f** Immunofluorescence of LTF, (*red*) and neutrophil elastase, ELANE (*green*) in a mouse brain at hematoma-affected area at 24 h after ICH. The nuclei were labeled with DAPI (*blue*). *Scale bar* = 50 μm. The insertion window highlighted the co-localization of LTF, with ELANE⁺ neutrophil. **g** LTF, protein in the ICH-affected rat brains (harvested from animals perfused with saline to flash intravascular cells), in the same animals as in **e**, and established using Western blot (*n* = 5)

Using immunohistochemistry (Fig. 5d) and enzyme-linked immunosorbent assay (ELISA) (Fig. 5e), we confirmed that LTF is an abundant component of mature circulating PMNs and that ICH (an insult that increases blood IL-27; see Fig. 1) increased LTF content in the peripheral blood PMNs (Fig. 5e; *middle bar*), and that rIL-27 treatment after ICH further augmented PMNs' LTF content (Fig. 5d; *red bar*).

Importantly for ICH pathogenesis, using immunohistochemistry we detected LTF-positive PMNs in ICH-affected brain (Fig. 5f). We also found LTF to be elevated in the ICH-affected brains of animals treated with rIL-27 (Fig. 5g), despite the fact that neither neurons, astrocytes, oligodendroglia nor microglia (analyzed for mRNA expression; data not included), nor other immune cells (Fig. 5b) express LTF. These data suggest that PMNs may deliver LTF to the injured brain.

**LTF reduces damage in models of ICH**. Our final set of experiments sought to establish the relevance of LTF as a treatment for ICH. Since LTF is highly effective in the sequestration of iron, and iron-mediated toxicity contributes to ICH-induced damage, we evaluated the therapeutic efficacy of LTF using an in vitro ICH-like injury model and clinically relevant ICH animal models[34–36].

We first employed an in vitro model, based on exposing neuron-glia co-culture to RBC lysates. Using this model, we showed that RBC lysates are toxic to neurons (Fig. 6a), and that presence of recombinant LTF in culture media improves neuronal survival in this injury model (Fig. 6b).

In the next experiment, we employed a well-validated ICH model (based on an intrastriatal autologous blood injection) in both rats (Fig. 6c–e) and mice (Fig. 6f, g)[34, 36]. First, in a proof of concept experiment, we treated rats with LTF at 30 min after the onset of ICH. These studies showed that LTF potently reduced NDS (Fig. 6c), brain edema (Fig. 6d), and oxidative stress (Fig. 6e). Second, using a mouse ICH model, we established that LTF has a very long, clinically relevant therapeutic window. Mice that received LTF as late as 24 h after ICH demonstrated a robust decrease in NDS (Fig. 6f) and improved clearance of iron from the hematoma-affected hemisphere (Fig. 6g).

**Discussion**
PMNs are the shortest-lived and most abundant cells of the innate immune system. Following ICH, PMNs enter the affected CNS within hours. Once there, they release various lytic enzymes and generate free radicals that may damage the surrounding brain tissue[15]. While PMNs are one of the first blood cell types to enter the injured brain, they continue entering damaged tissue for up to several weeks. This poses the question whether PMNs at the later stages of ICH pathogenesis are of the same damaging phenotype as these entering brain during the initial hours, and whether their phenotype could be altered to ameliorate their damaging capacity during recovery.

Our study is the first to demonstrate that IL-27 production increases within an hour after ICH in the brain and periphery, with increases seen in the ICH-affected brain (primarily from microglia), CSF, spleen, and peripheral blood. We found that IL-

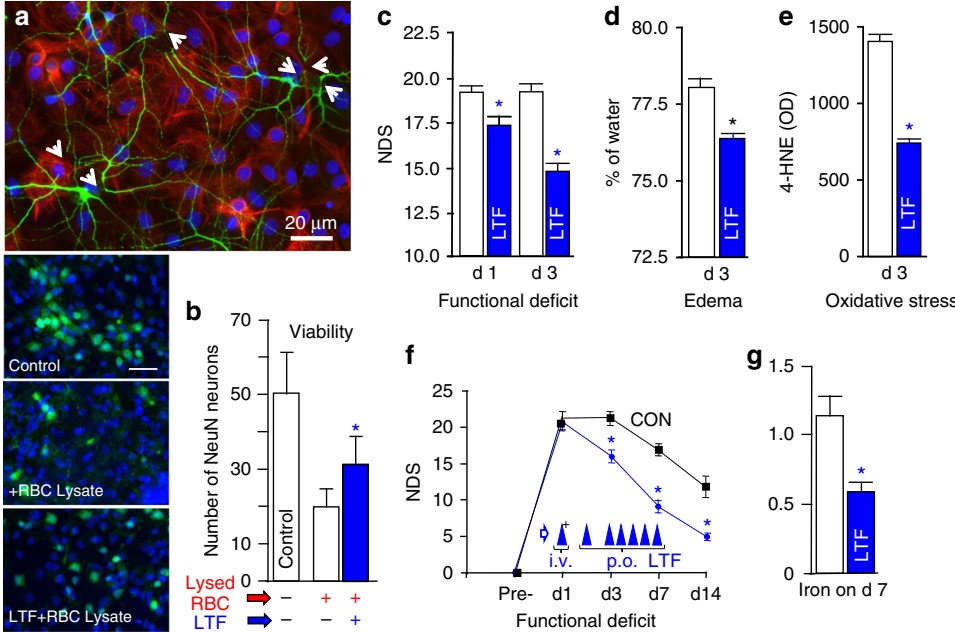

**Fig. 6** Neuroprotective effect of recombinant LTF in vitro (**a, b**) and in a clinically relevant rat (**c–e**) and mouse (**f, g**) model of ICH. **a** Representative immunofluorescence of MAP2[+]-neurons (*green*) and GFAP[+]-astrocytes (*red*) in rat brain cortical neuron-glial co-cultures at 24 h after exposure to RBC lysate (an ICH-like injury in vitro), demonstrating that lysed RBCs are toxic to neurons. *Arrows* show the swollen soma and broken dendrites. Astrocytes are less prone to the injury. **b** Representative images (*left* part of the panel) showing density (viability index) of NeuN[+]-neurons (*green*) in the rat neuron-glial co-cultures at 24 h after the exposure to lysed RBC in the presence or absence of rLTF, (5 µg/ml), which was added to the co-culture at 30 min prior to the insult (adding lysed RBCs). The nuclei were labeled with DAPI (*blue*). *Scale bar* = 50 µm. And a bar graph of NeuN[+]-neuron numbers at a 10× field. Data were mean ± SEM (*n* = 15). *$p \leq 0.05$ from all other groups; established using one-way ANOVA followed by Newman–Keuls post-test. **c–e** rLTF reduces ICH-mediated brain damage in rats after ICH. Rats were treated with rLTF (5 mg/kg, i.v. at 30 min after ICH plus 2.5 mg/kg p.o., at 24 and 48 h) and assessed for **c** NDS (a composite score for footfault, circling, wire, postural flexing, and forward placing test) at days 1 and 3 (*n* = 10), **d** oxidative burden (4-HNE; index of lipid peroxidation) at day 3 (*n* = 5), and **e** brain edema (% of water content) at day 3 (*n* = 5). **f, g** Recombinant mouse LTF, improves outcome after ICH in mice that were treated as late as 24 h after ICH (5 mg/kg, i.v. at 24 h after ICH plus 2.5 mg/kg p.o., on day 2–7) as determined with assessing **f** NDS on day 1, 3, 7, and 14 (*n* = 11) and **g** iron clearance from the hematoma-affected brain on day 7 after ICH (*n* = 7). All data were presented as mean ± SEM. *$p \leq 0.05$, compared with the vehicle control. The *p*-values in **d, e, g** were established using paired *t*-test; in **c, f**, the longitudinal data analysis was performed and corresponding *p*-values were provided using the mixed effects model

27 acts upon maturing BM-PMNs (in vitro and in vivo) to modify their phenotype, resulting in the induction of PMN components such as Hp and LTF that could play beneficial or protective functions in ICH while reducing the expression of potentially harmful pro-inflammatory iNOS, MMP9, or NOX2. Most importantly, we found that these effects of IL-27 treatment coincide with its therapeutic benefit in an ICH model, including reduced brain edema and an improvement in acute and sub-acute functional outcomes. Furthermore, we have also shown for the first time that LTF, a protein highly expressed in PMNs, is significantly increased in BM-PMNs, circulating PMNs, and the ICH-affected brain in response to the treatment with rIL-27. Most importantly, direct treatment with LTF produces a similar therapeutic effect to IL-27 treatment in our ICH model.

PMNs are short-lived myeloid cells that originate and mature in the BM[15]. Most PMN constituents are synthesized at various stages of their development before storage in various intracellular granules[15]; and mature PMNs entering circulation act as shuttles to deliver these constituents to the site of infection or tissue damage[16–19]. After degranulation (release of the granule content), circulating PMNs retain only a partial capacity to renew these constituents, indicating that the biological properties of PMNs at the site of their infiltration may depend on their phenotype (the chemical composition of their granules) that was pre-established in the BM. Thus, based on the outcome of our present study, it is likely that IL-27 generated in response to ICH could signal through the IL-27R[30] on maturing PMNs in the BM

to modify PMN phenotype to a less neurotoxic (or even somewhat beneficial) type. Indeed, depletion of PMNs prior to ICH mitigates ICH-induced damage[12], while post-ICH depletion of PMNs has a neutral effect (as seen in our present study), suggesting a potential temporal difference in the function that PMNs play in ICH pathogenesis. Finally, our results suggest that IL-27 is less effective in reducing ICH-mediated damage in PMN-depleted animals, implying that IL-27-modified PMNs entering the ICH-affected brain during the sub-acute phase may cause less damage and possibly augment the recovery process. Furthermore, our study indicates that BM-PMNs exposed to IL-27 upregulate the IL-27R itself, suggesting that IL-27 acts in a feed-forward mechanism to further amplify and strengthen the anti-inflammatory phenotype of PMNs.

The mechanisms responsible for the increased IL-27 level in the blood and CSF after ICH has not been studied in this project; however, it is possible that IL-27 generated by the ICH-activated microglia could enter the peripheral circulation and or CSF and contribute to its elevated levels. Furthermore, we have observed that after ICH, expression of IL-27 is also elevated in the spleen. While we do not know the routes of communication between ICH-affected brain and spleen, the earlier research demonstrated that chemical (e.g., damage-associated molecular pattern molecules, such as HMGB1[37]) and/or electrical (efferent neurogenic polysynaptic relay through the vagus nerve[38]) signals originating from the injured brain could modulate the spleen activities, that could also include IL-27 production.

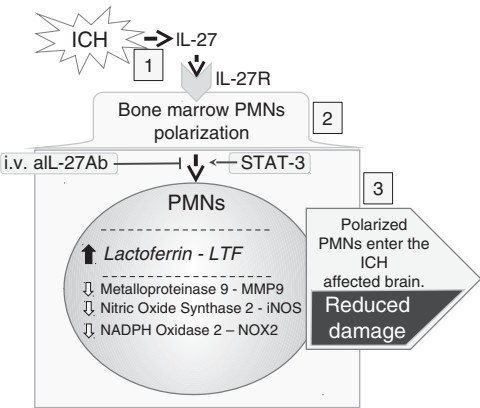

**Fig. 7** In response to injury caused by ICH, activated brain microglia and spleen rapidly produce IL-27 (**1**), a cytokine, which modifies the developing PMNs in the BM to express more of a potent iron-sequestrating LTF and less of pro-inflammatory factors (**2**). This process is referred to as PMNs polarization. In consequence, these polarized PMNs may carry more cytoprotective LTF. Upon entering the injured brain, the PMNs may release more LTF that through sequestrating the cell-free ferric iron (produced due to intra-hematoma hemolysis) alleviates oxidative stress, ICH-deposited iron, and ultimately neurological damage

While the effect of IL-27 on PMNs is likely important, the therapeutic effect of IL-27 in ICH may also include the regulation of various components of the immune system, including B, T, and NK cells[15, 24, 25]. This is especially important, as various immune cells are implicated in ICH pathobiology[39, 40]. However, we have also demonstrated here that LTF (a protein whose main systemic source is circulating PMNs) is upregulated in PMNs in response to IL-27 and that LTF, when used therapeutically, protects from ICH-induced brain edema and neural tissue damage. In addition to LTF, IL-27 enhanced the expression of Hp, a key acute phase hemoglobin-detoxifying protein[36]. In our earlier work, we showed that Hp plays an essential function in protecting the brain from blood toxicity after ICH[36]. Taken together, this evidence argues that PMNs may augment the detoxification of cytotoxic hemoglobin and iron, two components associated with secondary brain injury after ICH[1, 7].

In addition to upregulating the beneficial components of PMNs, IL-27 reduced the expression of NOX2, MMP-9, and iNOS, enzymes that could aggravate ICH pathogenesis. NOX2 is a key constituent of NADPH oxidase[41], which is highly abundant in PMNs and is a main source of PMN-generated superoxide and oxidative damage at the site of ICH-induced inflammation. Genetic disruption of NADPH oxidase has been demonstrated to protect animals from ICH-mediated damage[42]. The inducible form of NOS (iNOS) is highly expressed in PMNs and was demonstrated to contribute to brain tissue damage following ischemic stroke[43]. As ICH triggers robust PMN recruitment into the ICH-affected brain parenchyma[11], amelioration of iNOS expression in PMNs with IL-27 may have a protective effect. PMN-derived MMP-9 contributes to extracellular matrix and basal lamina degradation, blood-brain-barrier disruption, and hemorrhagic transformation[44, 45]. MMP-9 inhibition was shown to effectively ameliorate damage associated with ICH[46]. However, MMP-9 is proposed to act as important contributor to post-stroke remodeling and repair at later stage[47]. It is interesting to note that IL-27 caused only partial inhibition of MMP-9 expression, suggesting it may play a limited function in the improved functional outcomes seen in this study. However, further studies to address these issues are warranted.

One of the most promising findings of this study is the identification of LTF, a pleotropic iron-binding glycoprotein from the transferrin family, as a potential innate defense mechanism against damage caused by ICH. LTF is a well-conserved 80 kDa protein found in mucosal secretions as a product of epithelial cells and in the specific granules of PMNs[48]. LTF is generally recognized as a first-line defense molecule in protection against microbial infections due to its ability to sequester iron away from invading microbes[49]. One molecule of LTF can bind two iron atoms with the extraordinary high affinity (Kd~$10^{-22}$ M). Unlike other iron binding proteins, LTF retains iron molecules even at low pH[50]. This property is important in sequestering free iron to form iron–LTF complexes in inflamed tissue (acidic pH), preventing free $Fe^{2+}$ engagement in Fenton's reaction which produces oxidative damage and inflammation[7]. This is particularly important in ICH pathogenesis, as the RBC in the ICH hematoma undergo hemolysis, generating large quantities of hemoglobin, heme and iron, leading to oxidative damage to brain cells and the neurovasculature[1, 7, 8, 36, 51]. Thus, iron neutralization by LTF may represent an essential step in the removal of free iron and the reduction of iron-mediated oxidative damage. Since PMNs are a key source of peripheral LTF, and PMNs can carry LTF to the ICH-affected area, we postulate that PMNs play an essential function in mitigating iron-mediated oxidative damage to brain tissue after ICH. Furthermore, our data indicate that approaches that augment LTF content in PMNs represent promising therapeutic strategies for improved outcomes after ICH.

In contrast to most experimental and clinical studies to date, which used various types of LTF not fully compatible for human parenteral administration, in the present study we employed GLP quality recombinant LTF expressed based on the wild-type sequence in eukaryotic cells. This approach guarantees that the LTF used in our study is correctly glycosylated and possess optimal bioactivities[52]. And LTF robustly improved iron removal and reduced oxidative stress, brain edema, and neurologic dysfunction, suggesting that direct LTF supplementation represents a viable candidate for ICH treatment. Since LTF can penetrate the BBB[53], we believe that parenteral LTF administration could directly improve iron neutralization at the site of ICH. Most importantly, our studies demonstrate a robust protective effect of LTF when given early after ICH or even 24 h after ICH, indicating a uniquely long therapeutic window.

In conclusion, our studies demonstrate the following: (1) in response to ICH, IL-27 is produced in the brain (with microglia serving as the primary brain cell source) and also in the spleen; (2) IL-27 acts on developing PMNs in BM to reprogram the expression of various genes known for their strong relevance to ICH pathogenesis, while (3) improving ICH outcome, reducing edema and augmenting iron and hemoglobin clearance from the ICH-affected brain; (4) neutralization of IL-27 after ICH aggravates the injury; (5) IL-27-mediated protection is reduced in PMN-depleted animals; (6) IL-27 augments the production of LTF in PMNs; and (7) LTF used as a therapeutic agent robustly reduces the damage caused by ICH with 24 h therapeutic window. We propose that LTF released from PMNs entering the ICH-affected brain may neutralize iron toxicity, and that IL-27 may augment this process (Fig. 7). This study also shows a novel function of IL-27 in PMN polarization based on reciprocal communication between brain and BM. Therefore, modulation of IL-27 and LTF and their associated pathways may represent novel, powerful targets for the treatment of ICH.

## Methods

All animal studies followed the guidelines outlined in *Guide for the Care and Use of Laboratory Animals* from the National Institutes of Health and were approved by the Animal Welfare Committee of The University of Texas Health Science Center

at Houston. All studies were performed using a randomization (coin toss) approach and all analyses were performed by investigators blinded to treatment assignments (animals were coded for the group allocation). Animals were kept in a specific pathogen-free/SPF environment and were fed a standard mouse/rat diet, and housed in standard mouse/rat cages on a 12-h inverted light–dark cycle. Behavioral analyses were carried out from the hours of 10:00 AM to 4:00 PM.

**ICH in rat and mice**. ICH in rat and mouse was induced by intra-striatal injection of autologous blood as described previously[6, 34, 35]. Briefly, male Sprague Dawley (SD) rats (250–350 g) or male C57BJ6 mice (25–30 g) under chloral hydrate anesthesia (0.35 g/kg; i.p.) were immobilized onto a stereotaxic frame. A 1-mm-diameter burr hole was drilled in the skull and a 26-gauge stainless steel cannula was inserted for blood infusion (collected from femoral artery; 15 μl/5 min for mice, or 35 μl/5 min for rats) into the left corpus striatum (for SD rat, 0.5 mm anterior to bregma and 4.0 mm lateral to midline, and 5.6 mm deep to skull; for mouse, 0.0 mm anterior to bregma and 3.0 mm lateral to midline, and 3.5 mm deep to skull). Core body temperature was maintained at $37 \pm 0.5\,^{\circ}C$ during entire surgery and for 2 h afterward.

**Tissue harvesting**. Animals were anesthetized with chloral hydrate (0.5 g/kg; i.p.) and intracardially perfused with ice-cold PBS. Subsequently, for histology or bio-chemical analyses, the spleen, whole brains, or the sub-dissected tissues repre-senting the area of sub-cortical striatum affected by ICH were snap frozen by submersion in $-80\,^{\circ}C$ 2-methylbutane and stored in $-80\,^{\circ}C$ freezer prior to cryosectioning, RNA isolation, or protein analyses.

**Blood preparation and cerebrospinal fluid harvesting**. Blood serum or plasma from rat or mouse was obtained via cardiac puncture. Freshly drawn whole blood (500 μl–1 ml) in a micro-centrifuge tube was allowed to clot (30 min–2 h after harvesting at 4 °C) before centrifugation at $1000 \times g$ for 10 min (4 °C)[54]. The supernatant (serum) was transferred to another micro-centrifuge tube and stored in $-80\,^{\circ}C$ freezer until being used for ELISA assay. To harvest plasma, whole blood from cardiac puncture was collected in EDTA-coated tube (BD Vacutainer, K2 EDTA) before centrifugation at $1000 \times g$ for 10 min. The supernatant was used for ELISA assay. The cerebrospinal fluid (50–100 μl) from SD rats was collected from the cisterna magna. After centrifugation ($400 \times g$ for 10 min at 4 °C), the super-natant were transferred to another micro centrifuge tube and stored in $-80\,^{\circ}C$ freezer until being used for ELISA assay.

**IL-27 ELISA**. The amount of IL-27 in rat microglial culture medium and IL-27 in CSF after ICH was measured using a rat IL-27 p28 ELISA kit (MBS2601322, My BioSource). The samples were diluted at 1:5 (for CSF) or 1:2 (for culture medium) with the Standard Diluent provided in the kit and then measured against the standard at 0–500 pg/ml, according to the manufacturer's instructions. The IL-27 p28 in mouse serum or plasma was measured using mouse IL-27 p28 ELISA kit (MBS176517, My Bioscience) and multiplex MAP mouse Th17 Magnetic Bead kit (MT17MAG47K-PX25, Millipore). The serum were first diluted in the Standard Diluent provided in the ELISA kit and then measured against the standard at 0–250 pg/ml, according to the manufacturer's instructions.

**Primary cortical neuron cultures**. The cortices of E-18 pre-natal embryos were dissected and dissociated by triturating as we previously described[55], with some modification. The dissociated cells were plated on poly-L-lysine-coated culture plates in Neurobasal medium with B27 at 300–700 cells/mm$^2$. The cells were maintained in a $CO_2$ incubator (5% $CO_2$, 21% $O_2$) at $37.0 \pm 0.5\,^{\circ}C$. Half of the culture medium was changed every 3 days. After a total of 9–12 days in culture, the neuronal cells formed extensive axonal and dendritic networks and were ready for the experiments. Using MAP2 immunofluorescence, we confirmed that ≥90% living cells were MAP2$^+$ neurons.

**Astrocyte/oligodendrocyte/microglia cultures**. The individual primary cortical glial cell cultures were isolated and purified from rat brain co-cultures using p1-p2 pups, as we previously described[6]. First, the cells from brain cortices were seeded in 75 cm$^2$ TC flasks and cultured for 14 days. The loosely adherent microglia were harvested from the culture medium after slight shaking. After centrifugation at $400 \times g$ for 5 min, microglia were collected and re-plated onto poly-L-lysine-coated TC plates, with or without 12-mm diameter German-glass, at a density of $1–4 \times 10^5$ cells/ml. After removing the microglia, the co-culture flasks were continuously shaken at a speed of 220 rpm for 30 min to remove the other loosely attached cells including microglia and proliferating cells (cells at the proliferating stage are loosely attached to the astrocyte layer, and thereby relatively easily shaken off). The co-culture flasks were then changed into fresh culture medium and agitated at 220 rpm for 16–20 h to harvest the oligodendrocytes. After the above shaking steps, the tightly attached cell layer on the tissue culture flasks consisted primarily of astrocytes. Using immunostaining for microglia (CD68, MCA341, Serotec), oli-godendrocytes (OSP, Ab7474, Abcam), and astrocytes (GFAP, G3893, Sigma-Aldrich), we confirmed that microglia cultures 24 h after re-plating were

≥ 96% pure; oligodendrocyte-rich cultures 10 days after re-plating were ≥ 84% pure; and the astrocyte-rich cultures 10 days after re-plating were ≥ 90% pure.

**LTF treatment of co-cultured cells**. For the cell culture experiments, 5 μg/ml recombinant human lactoferrin (rLTF) was applied to the cortical neuron-glia co-culture medium at 30 min before adding lysed RBC, and continuously incubated for 24 h after incubating in RBS lysate. The cells were then fixed for MAP2-immunofluorescence, GFAP-immunofluorescence, or NeuN-immunofluorescence labeling and cell counting.

**Blood PMNs isolation**. Peripheral blood was drawn via cardiac puncture and blood PMNs were purified by Ficoll-Paque gradient centrifugation as described[56]. Mouse PMNs were further purified using Anti-Ly-6G MicroBeads kit (Miltenyi Biotec) and EasySep$^{TM}$ mouse Neutrophil Enrichment kit (STEMCELL$^{TM}$ Technologies).

**LTF ELISA**. LTF levels in blood serum and blood-derived PMN lysates were measured using a rat LTF ELISA kit (MBS728256, MyBioSource.com).

**BM-precursor PMNs**. The precursors of PMNs (metamyelocytes, myelocytes, promyelocytes, and myeloblasts) present in BM were harvested using gradient centrifugation as previously described, with minor modifications[57, 58]. Briefly, rat or mouse BM was harvested from femurs and tibias of both hind legs by flushing with ice-cold $Ca^{++}/Mg^{++}$-free Hank's balanced salt solution (HBSS) using a 1 ml syringe (22G needle). After centrifugation at $400 \times g$ for 5 min, the RBCs in the cell pellet were removed with RBC Lysis Buffer and the cells were rinsed with HBSS. Next, the cell pellet was re-suspend in 3–6 ml 45% Percoll Solution and the cells were filtered through a cell strainer of 100 μm porosity to eliminate eventual cell clumps and fat aggregates. The cells were loaded on top of a discontinuous Percoll density gradient (from the bottom of a 15 ml falcon tube, a sequence of 81% (3 ml), 62% (2 ml), 55% (2 ml), and 50% (2 ml) Percoll solutions). After centrifugation at $700 \times g$ for 30 min (without the brake), we collected the cells from the interface of the 62/81% (mature PMN fraction) and also from 62/55% (immature-to-mature PMN), and 55/50% (immature PMN), and washed them with HBSS before culture or other use.

**RBC isolation**. Rat blood RBC were prepared using column density gradient centrifugation (BD Vacutainer® CPT™) as reported[6].

**Flow cytometry**. Blood was collected by cardiac puncture into tubes anti-coagulated with 4% citric acid. Two femurs from each mouse were dissected out and flushed with 10 ml of RPMI1640 culture medium. RBC in blood and BM were lysed with an ammonium chloride solution (Stem Cell Technologies). Cell number was quantified on a BD Countess Cell Counter and all samples were diluted to a concentration of 500,000 cells/ml. Blood and BM leukocytes were washed, stained for viability, and blocked with mouse Fc Receptor Blocking Solution fluorophores: CD45-vf450, CD11b$^-$APC-Cy7, and Ly6G-PE, Ly6C-APC, CD3-Pe-Cy7. For intracellular staining for LTF, cells were incubated in a fixation/permeabilization buffer (BD Biosciences) for 20 min, then washed and incubated with a rabbit anti-mouse lactoferrin primary antibody (L-3262, Sigma-Aldrich) or rabbit anti-mouse control IgG for 30 min. After washing, cells were incubated with a goat anti-rabbit secondary conjugated to AlexaFluor488 (Invitrogen) for 30 min. For full details on all targets, antibody clones, sources and dilutions for antibodies, and staining reagents used, please see Supplementary Table 3. Data were acquired on a Cytoflex S cytometer (Beckmann Coulter) and analyzed by FlowJo Software (Tree Star). PMNs were identified as CD45$^+$CD11b$^+$Ly6C$^{Int}$Ly6G$^+$ cells, inflammatory monocytes were identified as CD45$^+$CD11b$^+$Ly6C$^{High}$Ly6G$^-$ cells, and T cells were identified as CD45$^+$CD11b$^-$CD3$^+$ cells. Fluorescence minus one controls and/or isotype controls were used to determine positive gates for each antibody. The MFI of LTF was quantified within the PMN population, with T cells and inflammatory monocytes serving as negative controls for LTF fluorescence. For full details regarding the flow cytometry gating strategy for Figs. 3e and 5c, please see Sup-plementary Fig. 1.

**Reverse transcription-polymerase chain reaction (RT-PCR)**. The ipsilateral striatum surrounding the hematoma was dissected on ice, snap-frozen, and pro-cessed for mRNA extraction using Trizol-Reagents. The primary brain cells or purified BM-PMNs in culture were harvested, washed with PBS, and lysed in Trizol-Reagent. RT-PCR analyses were done as described previously[6, 59]. GAPDH was used as an internal standard. The sequences of primers are listed in Supple-mentary Table 1. Measurements of the gene products were normalized to the optical density of GAPDH bands. The results were calculated as optical density or percentage change over the control (naïve animal in vivo; media control in vitro).

**Western blot**. Protein levels were determined using Western blotting, as pre-viously described[6]. Rabbit anti-lactoferrin (bs-5810R, Bioss), mouse anti-STAT3 (9139 Cell Signaling), rabbit anti-p-STAT3 (9131S, Cell Signaling), or chicken anti-

GAPDH (Millipore, AB2302) immunopositive bands were visualized using goat anti-rabbit IgG-HRP (invitrogen) or goat anti-chicken IgG-HRP (Invitrogen) and ECL (Pierce, Rockford, IL) (see Supplementary Table 2 for more information). Semi-quantification of luminescence signal intensity on X-ray film was determined by the analyses of optical density, using a Computer-Assisted Kodak Analysis (EDAS) 290 system. Western blots of Fig. 2g (STAT3) and Fig. 5g (LTF) are included as Supplementary Figs. 2 and 3, respectively.

**Immunofluorescence**. Immunohistochemistry for rat or mouse lactoferrin, IL-27 p28, PMNs (ELANE), MAP2, NeuN, and GFAP was performed using the procedure as described[6]. Briefly, cells grown on German Glass, or smears of blood on glass slides, or spots of BM-PMNs on glass slides and coronal brain sections were treated with 95% methanol containing 5% acetic acid for 10 min at −20 °C and then incubated with primary antibodies (Supplementary Table 2) overnight at 4 °C. Goat anti-mouse, anti-rabbit, or anti-rat IgG conjugated with Alexa Fluor 488 or 546 (Invitrogen, USA) was used to visualize the fluorescent signals. The nuclei were visualized with DAPI. The immunofluorescence signals were observed under a Zeiss Axioskop 2 fluorescence microscope, which was equipped with CCD camera and operated by MetaMorph 7.4 software. The images were acquired using the filter sets at Ex/Em of 490/520 nm for Alexa Fluor 488, Ex/Em of 550/575 nm for Alexa Fluor 546, and Ex/Em of 365/480 nm for DAPI-labeled nuclei.

**PMNs counting**. The number of PMNs from each brain section at the needle insertion site is calculated. PMN cell counting in mouse peripheral blood was performed with tail blood smears on slides after immunofluorescent labeling with a rat anti-PMN antibody (ELANE, ab53457, Abcam). The digitized images (4 × 4 images under 20× lens) were acquired and stitched and the number of ELANE$^+$-neutrophils was automatically counted with CellSens software (Olympus).

**NDS measurement**. All behavioral tests in mice and rats were conducted in a quiet and low-lit room by an experimenter blinded with respect to the treatment groups. Pre-tests were done to exclude abnormal animals. Only animals with less than 20% foot-faults and animals that had no preference in forelimb placing were subjected to ICH. An individual test score and a combination test score from a battery of behavioral tests (Foot-fault, Forelimb Placing, Postural Flexing, Wire and Circling) was used to measure the NDS, as we reported earlier[6, 60]. Grand composite NDS combined all the tests assumed an equal weight of each of the tests.

**IL-27/anti-IL-27 Ab and LTF administration**. Recombinant mouse IL-27 (rIL-27, 577404, Biolegend, a heterodimer of mouse recombinant IL-27 EBI3 and p28 linked by a GGGSGGGSGGGTGGGS linker) at 50 ng/kg was administered i.v. at 0.5, 24, and 42 h after ICH for the experiments probing 48 and 72 h outcomes. In the sub-acute study, rIL-27 (50 ng/kg) was administered i.v. at 30 min and then daily s.c., day 1 to 6 after ICH. To assess the therapeutic relevance of IL-27, we treated mice with rIL-27 (as above) or with mouse anti-IL-27 antibody (αIL-27; Biolegend, 516912) at 40 μg/mouse i.v. at 0.5, 24, and 42 h after ICH to neutralize endogenous IL-27, with mice injected with rat IgG1 (400427, Biolegend) serving as a control. We measured the LTF content in mature blood PMNs and expression of selected genes in BM-PMNs at 48 h with LTF ELISA, the LTF content in mature blood PMNs at 72 h with flow cytometry, and the NDS at 72 h and 7 days. For in vivo LTF experiments, rLTF (gift from PharmaReview Corporation, Houston, TX) was dissolved in saline and administered at 5 mg/kg, starting 30 min after ICH, plus 2.5 mg/kg, p.o. on day 1 and day 2 after ICH, with outcome measurements on day 3 in the acute treatment experiments. Saline was used as the vehicle control. For the sub-acute study, recombinant mouse LTF (gift from PharmaReview Corporation, Houston, TX) was dissolved in saline and administered 5 mg/kg, i.v. at 24 h after ICH plus 2.5 mg/kg, p.o., on days 2–7. Saline was used as the vehicle control. Outcomes were measured on day 1, 3, 7, and 14.

**PMNs depletion in mice after ICH**. To understand the function of PMNs in rIL-27-induced protection (Biolegend, 577404, a heterodimer of mouse recombinant IL-27 EBI3 and p28 linked by a GGGSGGGSGGGTGGGS linker), we injected rat monoclonal α-Ly-6G, a PMN depleting antibody (BioXCell, 1A8, BE0075) at 500 μg/mouse at 2, 24, and 72 h after ICH (i.p.). A rat IgG 2a isotype control antibody (BioXCell, 2A3, BE0089) was used as the control. We measured PMN numbers in the peripheral blood to confirm the efficacy of PMN depletion. NDS was measured on day 1 and day 3 after ICH.

**Hematoma size measurement**. Hematoma resolution was assessed by measuring the amount of hemoglobin remaining in the hematoma-affected brain on day 7 after ICH, as detailed previously[6]. Briefly, mice under chloral hydrate anesthesia were perfused with ice-cold PBS to remove intravascular hemoglobin. Intraparenchymal hemoglobin in the homogenized ipsilateral striatum was measured using Drabkin's reagent[6, 61]. Peripheral blood (0, 2, 4, 8, 16, or 20 μl) was added to the homogenate of naïve perfused brain and used to prepare a standard calibration curve. The data were expressed as blood volume per brain homogenate.

**Non-heme brain tissue iron determination**. To establish the location of iron in the brain sections, we used enhanced Perl's reaction[62]. Brain sections fixed with 2% formalin were incubated in Perl's solution (1:1, 5% potassium ferrocyanide and 5% HCl) for 45 min. After washing with distilled water, the sections were incubated in 0.5% diaminobenzidine tetrahydrochloride with nickel for 60 min.

To quantify non-heme iron in the ICH-affected brain, animals were perfused with saline before decapitation. A coronal 4-mm thick slice centered on the injection needle tract was generated, divided into ipsilateral and contralateral sides and weighed out. The tissue iron content was determined according to a method described by Weinfeld et al.[63] using an Iron Colorimetric Assay Kit (CFE-005, JaICA). Briefly, the brain tissue was homogenized in 1 ml of dH2O. After adding 1 ml of 8.5 M HCl the brain tissue samples were hydrolyzed at 90 °C for 60 min. After cooling, 2 ml of 20% trichloroacetic acid was added to precipitate proteins, and supernatant was collected after centrifugation. The supernatant was run through an acid-washed filter, and the precipitate was washed with 1 ml of 4.25 M HCl plus 20% trichloroacetic acid (1:1). The supernatant was collected, and 4 ml of 1 M sodium citrate was added. The pH was adjusted to 3.1, and the final volume was adjusted to 25 ml. The total non-heme iron content was assayed with Spectrometer at 560 nm against iron standards.

**Brain edema**. Brain edema was measured using the wet weight/dry weight method[36]. Briefly, the brains were removed without perfusion and the ICH-affected brain hemispheres were dissected. A brain coronal section (4-mm thick; 2 mm anterior and 2 mm posterior to the blood injection site) was excised. The tissue weight was determined before and after drying in a 95 °C oven for 48 h. Brain edema was expressed as percent of water content: (wet weight−dry weight)/wet weight × 100.

**Statistical analyses**. Data were expressed as a mean ± SEM. For in vitro experiments, we pooled samples from three culture wells and repeated the experiments three times. We performed statistical analyses using the GraphPad and InStat programs and SAS 9.4 (Cary, NC). One-way analysis of variance (ANOVA) followed by Newman–Keuls post-test was used for multiple comparisons. Paired $t$-test was used when two groups were compared. Two-way ANOVA followed by Bonferroni post-test to compare replicate means was used to compare NDS between two groups at different time points. Also, the longitudinal data analysis was performed using the mixed effects model.

**Data availability**. The data that support the findings of this study are available from the corresponding author on rational request.

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

## Acknowledgements

This work was supported by National Institute of Neurological Diseases and Stroke (NINDS), grants 5R01NS096308 and 2R42NS090650.

## Author contributions

X.Z. contributed to the design of the study, carried biochemical characterization of neutrophil in vitro and in vivo, and assisted in analysis of behavioral and biochemical data and writing the manuscript. C.-H.L. conducted initial studies with the cytoprotective effect of IL-27 in vitro. G.S. performed all the animal surgeries and treatments. M.K. provided human lactoferrin and advised on all the experimental design aspects involving lactoferrin. S.-M.T. performed behavioral analyses. M.R.-O. conducted flow cytometry and assisted in writing the manuscript. J.A. initiated the project, conducted data analyses and provided overall guidance, and wrote the manuscript.

## Additional information

**Competing interests:** Dr Kruzel is employed by PharmaReview Corporation, Houston, TX. The remaining authors declare no competing financial interests.

