## [Peer Review File · Nature Communications]

Reviewers' comments:

Reviewer #1 (IL27, inflammation/anti-inflammation) (Remarks to the Author):

In this submission the authors examine the impact of the cytokine IL-27 on a model of ICH and they link its protective effects (after treatment) with the presence of a neutrophil populations that expresses increased levels of lactoferrin and they suggest that its ability to scavenge Fe leads to an improved outcome. In this setting it implies that neutrophils are neuroprotective – but its not clear how this fits with a literature in which the depletion of neutrophils improves a functional outcome (see Sansing et al 2011) while in other studies IL-27 blockade results in increased neutrophil responses (Wirtz JEM). There may be ways to reconcile these studies – but it is not so obvious in the way the manuscript is structured. Certainly the authors need to be clearer about the impact of IL-27 treatment on other populations besides neutrophils. There is a limited literature on lymphocytes as a source of IL-17 during ICH (IL-17 can mobilize neutrophils) and IL-27 is certainly a potent inhibitor of IL-17 so might (might not) be relevant to know if this is relevant to the protective effects of IL-27.

One of the major take home messages is the idea that the protective effects of IL-27 are mediated through neutrophils (and lactoferrin). The data for this is good but somewhat indirect. A more direct way to test this idea is to determine if the protective effects of IL-27 treatment is ablated by the depletion of neutrophils. This is not a trivial request and the results are not predictable (see paper by Sansing) but are key to testing the model proposed.

IL-27 treatment promotes neutrophil expression of lactoferrin in vivo but no data are presented that show a direct effect of 27 on neutrophils. However, the literature is a bit mixed on this topic and it would be relevant for them to show that IL-27 activates phospho STAT3 or STAT1 in neutrophils. Even at day 1 there looks to be some effects of IL-27 treatment – so it is rapid. Can they use blood from IL-27R KO mice to see if the IL-27 impacts this blood derived population that migrates into the tissue. This is not a must do but would certainly help to define the cells that IL-27 affects. If the IL-27 no longer has any protective effects when the blood comes from a IL-27R ko into a wt mouse then that would suggest the effects are not mediated through bone marrow derived neutrophils.

Fig 3, the legend title is confusing. Here they use mice with ICH and compares the effects of anti-IL-27 versus IL-27 on a select panel of genes. While Fig 3 compares the therapeutic effect of IL-27 treatment they need to also show if blocking IL-27 alter outcome – it should make things worse associated with reduced neutrophil expression of lactoferrin. Even if that is not the result it may be explained by a difference between local production versus treatment wit a bolus and that would be fine,

Figure 6A –Arrows show the swollen soma and broken dendrites. In the current version there is no image to compare this with and it seems from panel B that 50% of the neurons are not viable even without adding lysed RBC. These data sets beg the question of what neuronal cell death looks like in vivo after ICH and whether treatment with IL-27 or lactoferrin alters neuronal survival.

Minor Points

The authors need to improve the referencing of the relevant literature. For example, when they discuss the unique ability of IL-27 to limit immune-mediated pathology and autoimmune responses - ref 19 is not appropriate. Similarly, I think they are missing references that show the impact of IL-27 on HSC (there are at least 3 in Blood and Plos Pathogens), the JEM paper by Wirtz that shows that neutralization of IL-27 in a model of CLIP leads to improved neutrophil responses. There is also a previous literature on the production of IL-27 by different glial cell populations during EAE.

In Fig 1 the authors need to be clear about what they are measuring. IL-27 is a heterodimer that contains EBi3 (the other chain) and the receptor is a heterodimer that includes gp130 the other receptor chain. The ELISA measure IL-27p28 not the heterodimer. In addition, while their data indicate that IL-27 is induced in the absence of studies they need studies that determine the impact of the endogenous IL-27.

Fig 2b (validation of microglial cultures by IFA) can be removed or be supplemental. Similar comment

for Fig 2e. In Fig 2 the authors use rat microglia and show they respond to RBC and produce pg amounts of IL-27. It is unclear in its current form if this is a lot of IL-27 and it would be good to have a positive control (LPS and/or type I IFNs) as a comparator.

For 5a/c there is no specific control to compare these with. In Fig 5b the authors stain for lactoferrin and conclude that IL-27 treatment is upregulating its expression in the mouse neutrophils. However, based on flow, modest differences in size and granularity can lead to these types of differences and it would be helpful to be able to compare the FSC and SSC plots for these different populations.

Data is plural.

Good literature that for some stimuli the induction of IL-27 is indirect – through type I IFNs but it is less clear how rbc would promote IL-27.

Reviewer #2 (IL27, hematopoiesis) (Remarks to the Author):

This manuscript by Zhao et al. describes the protective roles of IL-27 and lactoferrin (LTF) in intracerebral hemorrhage (ICH). Using the mouse ICH model established by intra-striatal injection of autologous blood, the authors initially demonstrated that the immunoregulatory cytokine IL-27 is upregulated both centrally and peripherally after ICH and that exogenously administered IL-27 acts on developing polymorphonuclear neutrophils (PMNs) in the bone marrow, increasing their production of LTF that could neutralize hematoma toxicity. Moreover, administration of LTF reduced edema, enhanced hematoma clearance and improved functional outcomes in rat and mouse ICH models. The authors concluded that the modulation of IL-27 and LTF and their associated pathways represent novel, powerful targets for the treatment of ICH.

The topic addressed in this manuscript is interesting, overall, the experiments are well controlled, and most of the data are convincing.

1. However, the major criticism of this manuscript is the lack of demonstration whether the protective effect of endogenous and exogenous IL-27 is mediated by LTF. Blockade experiments of LTF in ICH and ICH plus IL-27 would further strengthen the authors' conclusion.

2. To prove that PMNs deliver LTF to the injured brain, the authors used Western blot (Fig. 5e). However, FACS staining of LTF in the brain-infiltrating PMNs as shown in Fig. 5b' or immunohistochemical analysis would be better for it.

3. IL-27 consists of p28 and EBI3, and IL-27 receptor consists of IL-27Ra (WSX-1/TCCR) and gp130. Therefore, the authors are required to examine mRNA expression of both molecules in each case (Fig. 1 and 2). In addition, the authors need to clarify whether the ELISA kit can detect heterodimeric IL-27 or p28 alone.

4. In Fig. 2C, intact RBC was used. In contrast, in Fig. 6b, RBC lysate was used. The authors need more explanation on the difference for the readers to easily understand it. Is this because RBC membrane but not lysate is necessary for augmentation of IL-27 expression? If so, the authors may add discussion on the molecular mechanism whereby RBC augments IL-27 expression.

5. Lane 397, the "immunoblot" should be "immunostaining" or "FACS staining".

6. Table 1 is missing (lane 235).

7. In Fig. 5b', staining with isotype-matched control antibody is necessary.

Reviewer #3 (CNS path) (Remarks to the Author):

Zhao et al. demonstrate that, in a model of intracerebral hemorrhage (ICH), the immunomodulatory cytokine IL27 may have beneficial effects by promoting iron removal, and by reducing edema and oxidative stress, effects attributed largely to neutrophil-derived lactoferrin. The data lead to two main conclusions: first, neutrophils are not uniformly deleterious in the setting of ICH, second IL27 is a potential therapeutic target for this deadly form of acute stroke.

The paper is of interest since it has relevance for both the basic pathology of ICH and to its treatment using approaches to enhance IL27 or its molecular effectors on the brain.

I have the following suggestions for improvement:

1. Abstract: the results on using anti-neutrophil approaches refer to ischemic stroke and not ICH.
2. IL27 has also deleterious effects in some settings (Bosmann-Ward, *J Leuc Biol*, 2013) that should be mentioned upfront.
3. While neutrophils are a key cell in ICH, other bone marrow derived cells are also the target of IL27, e.g., macrophages (Hirase et al *AJP-Heart* 2013), and need to be mentioned and/or addressed.
4. Is IL27 produced in brain (3 hrs) and periphery (spleen 24 hrs) independently? If so what are the cells predominantly involved? The culture data on microglia-neuron-astrocytes are *in vitro*, which may not reflect the cell-type expression *in vivo*. The upregulation in spleen is of interest and the cellular localization of the expression would help pinpoint the peripheral IL27 sources. Addressing the relative contribution to the production of IL27 and outcome of ICH by CNS and BM-derived cells would also be of interest, but it may be outside the scope of the paper.
6. The data on increased IL27 expression with RBC addition in microglia is of interest, but raises the question as to whether RBC would also increase expression in neurons and astrocytes.
7. The experiments with neutralizing antibodies are of great interest because they suggest a role of endogenous IL27 in the outcome of ICH. However, one wonders if neutralizing IL27 worsens the outcome of ICH.
8. Lactoferrin upregulation is a reasonable explanation for the reduction in injury afforded by IL27, but it would be helpful to assess the cellular expression of this protein in brain after ICH.
9. Is the protective effect of recombinant lactoferrin dose related?
10. The pitfalls of the model of ICH used should be mentioned. Data in another model would enhance the appeal of the paper.

REVIEWERS' COMMENTS:

Reviewer #1 (Remarks to the Author):

The authors have done a good job of responding to many aspects of the previous critiques. They have performed additional experiments and clarified the manuscript where appropriate. Its a timely study and a neat message about the role of IL-27 in this particular model and does set the stage for more translational studies and I look forward to seeing this develop.

The only very minor point - this is not a gotcha – but can the authors check the isotype of the Biolegend anti-IL-27 Ab? I think it's a mouse IgG2a and they are using a Rat IgG as an isotype control. Probably not a big deal – but this should be clear in the materials and methods.

Reviewer #2 (Remarks to the Author):

Since the authors have sufficiently addressed our concerns, I feel that this manuscript is now worthy of publication in the Nature Communication.

Reviewer #3 (Remarks to the Author):

The authors may want to mention in the discussion the potential mechanisms by which brain injury leads to increases in IL27 in blood and systemic organs. Neurogenic and neurohumoral systems have been described, linking the brain to systemic immunity (J Neurosci. 2015; 35:583; Immunol Rev. 2012; 248:188) and a brief mention may be of interest to the readers.

Line 67: perhaps "...consistent with the finding...mitigates..." rather than "suggesting"

Line 84: abbreviate IL27 receptor here

Zhao et al.

Neutrophil Polarization as a Therapeutic Target for Intracerebral Hemorrhage: The Role of IL-27

Response letter:

Reviewer 1.

1. In this setting it implies that neutrophils are neuroprotective – but it is not clear how this fits with a literature in which the depletion of neutrophils improves a functional outcome (see Sansing et al 2011) while in other studies IL-27 blockade results in increased neutrophil responses (Wirtz JEM)¹.

We thank Reviewer 1 for their insightful comment and we apologize for not describing this issue more clearly. Yes, it is generally accepted, based on the results from experimental animal research (though not supported so far by clinical trial results) that the neutrophils contribute to early damage after stroke. Neutrophils enter the brain parenchyma almost immediately after the stroke, with ICH (a hemorrhagic form of stroke) resulting in a more robust neutrophil (PMN) infiltration than after ischemia. Indeed, Sansing et al, 2011 demonstrated that PMN depletion (with anti-Ly6G antibody) 12h prior to ICH reduced the laterality index in a cylinder test, suggesting that neutropenia prior to ICH could limit the brain injury after ICH. While scientifically interesting, this approach has limited therapeutic implication.

As requested by the Reviewer, we have now performed a similar experiment with anti-Ly6G-mediated PMNs depletion (using the same 1A8 clone and isotype control Ab as Sansing et al.); however, in our study we have employed a 2h post-treatment paradigm. We hypothesized that the second wave of PMN infiltration (e.g., 24h after ICH) would not have damaging consequences. Indeed, we found that delayed PMNs depletion (an approach depleting 44% of circulating PMNs) has no effect on neurological deficit caused by ICH. This likely implies that PMNs may have biphasic functions, with the first phase being more detrimental. Finally, in a separate experiment to address the role of IL-27, we have shown that rIL-27 (used at concentration that reduces ICH damage) has a more beneficial effect in animals that have intact (vs. depleted) PMNs (fig.4c/d).

In addition, we have now included a short discussion on this topic within the paper and included the Wirtz et al., as reference.

2. The authors need to be clearer about the impact of IL-27 treatment on other populations besides neutrophils. There is a limited literature on lymphocytes as a source of IL-17 during ICH (IL-17 can mobilize neutrophils) and IL-27 is certainly a potent inhibitor of IL-17 so might (might not) be relevant to know if this is relevant to the protective effects of IL-27. One of the major take home messages is the idea that the protective effects of IL-27 are mediated through neutrophils (and lactoferrin). The data for this is good but somewhat indirect. A more direct way to test this idea is to determine if the protective effects of IL-27 treatment is ablated by the depletion of neutrophils. This is not a trivial request and the results are not predictable (see paper by Sansing) but are key to testing the model proposed.

We apologize for being too neutrophil-centric in our background section. We have now incorporated a paragraph on IL-27 and other white blood cells, and additional data has been incorporated into figure 4.

We certainly agree with the Reviewer that more direct evidence linking IL-27 and neutrophils could be important. To achieve this goal (per Reviewer request), we have now included data

demonstrating that IL-27 is more effective in reducing neurological deficit caused by ICH in animals with intact neutrophils vs. animals with induced neutropenia, as mentioned above. We hope that this additional piece of evidence could be helpful in strengthening the link between IL-27 and the beneficial polarization of neutrophils after ICH.

3. IL-27 treatment promotes neutrophil expression of lactoferrin in vivo but no data are presented that show a direct effect of 27 on neutrophils. However, the literature is a bit mixed on this topic and it would be relevant for them to show that IL-27 activates phospho STAT3 or STAT1 in neutrophils.

In response to this insightful comment, we have now added Western Blot data demonstrating that rIL-27 can induce STAT3 phosphorylation in BM-PMNs (fig 3g, new).

4. Even at day 1 there looks to be some effects of IL-27 treatment – so it is rapid. Can they use blood from IL-27R KO mice to see if the IL-27 impacts this blood derived population that migrates into the tissue. This is not a must do but would certainly help to define the cells that IL-27 affects. If the IL-27 no longer has, any protective effects when the blood comes from a IL-27R ko into a wt mouse then that would suggest the effects are not mediated through bone marrow derived neutrophils.

Thanks for this excellent suggestion. However, we believe that the interpretation of the suggested experiment could be considerably difficult. As mentioned by the Reviewer, IL-27 has broad pleotropic effect and as such, it is reasonable to predict that it can affect ICH pathogenesis through various targets. As stated above and immediately below, we have now included new data showing that rIL-27 is more effecting in ameliorating ICH damage in animals with intact PMNs as compared to neutropenic animals. This new results provide additional strong evidence for a potential association between IL-27 and neutrophils in mediating beneficial effect.

5. Fig 3, the legend title is confusing. Here they use mice with ICH and compares the effects of anti-IL-27 versus IL-27 on a select panel of genes. While Fig 3 compares the therapeutic effect of IL-27 treatment they need to also show if blocking IL-27 alter outcome – it should make things worse associated with reduced neutrophil expression of lactoferrin. Even if that is not the result it may be explained by a difference between local production versus treatment with a bolus and that would be fine.

As recommended by the Reviewer, we have now performed additional experiments to establish whether the neutralization of IL-27 (the manipulation that we previously showed causes alteration in gene responses; fig 3a) can also adversely affect neurologic outcome after ICH. Indeed, as anticipated (and predicted by the Reviewer), our new results (fig.3c/d) showed that depletion of IL-27 with anti-IL27 antibody (given daily for 3 consecutive days after ICH and resulting in 81% depletion) aggravates the post ICH deficit, suggesting that increased IL-27 after ICH may represent a self-defense mechanism after ICH.

6. Figure 6A –Arrows show the swollen soma and broken dendrites. In the current version there is no image to compare this with and it seems from panel B that 50% of the neurons are not viable even without adding lysed RBC. These data sets beg the question of what neuronal cell death looks like in vivo after ICH and whether treatment with IL-27 or lactoferrin alters neuronal survival.

We apologize for the ambiguous nature of fig 6b. The number of neurons (NeuN+ cells) indicated on the Y-axis depict not a % of surviving cells, but the actual average number of cells seen within the visual fields. The viability is 100% in a RBC free cell culture wells. In order to make this

presentation clearer, we now included an additional panel (fig 6b) showing the representative NeuN (neuronal) staining illustrating the neuroprotective effect of LTF in RBC-lysate mediated cytotoxicity.

7. The authors need to improve the referencing of the relevant literature. For example, when they discuss the unique ability of IL-27 to limit immune-mediated pathology and autoimmune responses - ref 19 is not appropriate. Similarly, I think they are missing references that show the impact of IL-27 on HSC (there are at least 3 in Blood and Plos Pathogens), the JEM paper by Wirtz that shows that neutralization of IL-27 in a model of CLIP leads to improved neutrophil responses. There is also a previous literature on the production of IL-27 by different glial cell populations during EAE.

We thank the Reviewer for pointing at the important references we left out in our previous version. (1) We have removed ref 19 as pertaining to immunoregulation, as this reference correctly belongs to a sentence right below. (2) As suggested, in the introduction, we have now included 2 references regarding the effect of IL-27 on HSC^{2,3} and thus indirectly on PMNs. (3) A reference pertaining to IL-27 neutralization, reactive oxygen species, and IL-27 receptor on PMNs by Wirtz et al.¹ was also added. Finally, in the introduction, we have include a reference indicating that previous research has suggested that IL-27 may also be produced by the astroglia⁴. Regarding the production of IL-27 by other glial cells, we further tested IL-27/p28 and EB13 mRNA expression in other brain cell types besides microglia. Upon exposure to RBC, only microglia upregulate IL-27 (see fig 2b).

8. In Fig 1 the authors need to be clear about what they are measuring. IL-27 is a heterodimer that contains EB13 (the other chain) and the receptor is a heterodimer that includes gp130 the other receptor is induced in the absence of studies they need studies that determine the impact of the endogenous IL-27.

Thank you for this helpful comment. Originally, we had characterized the expression of IL-27 p28 and IL-27Ra (WSX1). We have now added new data showing the expression pattern of EB13 (fig 1) and gp130 (fig 2). This data is included in Fig.1a and 1d (expression in brain and spleen), and in Fig 2a (expression in astrocytes, oligodendrocytes, microglia, and neurons).

9. Fig 2b (validation of microglial cultures by IFA) can be removed or be supplemental. Similar comment for Fig 2e.

We removed both figures.

10. In Fig 2 the authors use rat microglia and show they respond to RBC and produce pg amounts of IL-27. It is unclear in its current form if this is a lot of IL-27 and it would be good to have a positive control (LPS and/or type I IFNs) as a comparator.

Excellent advice! Indeed, our data could not answer how robust IL-27 induction is in response to RBC. The literature suggests that in macrophages, LPS triggers IL-27 expression that is highly robust and about 10 times stronger than INF γ ⁵. Thus, we have now compared induction of IL-27 in microglia by RBC, LPS, and IFN γ . This experiment demonstrates (now fig 2c) that RBC is more potent than either LPS or IFN γ in induction of IL-27 mRNA expression (checked by RT-PCR; data not included) and its secretion, as established by IL-27/28 ELISA (fig.2c).

11. In Fig 5b the authors stain for lactoferrin and conclude that IL-27 treatment is upregulating its expression in the mouse neutrophils. However, based on flow, modest differences in size and granularity can lead to these types of differences and it would be helpful to be able to compare the FSC and SSC plots for these different populations.

We do agree that differences in size (FSC) and granularity (SSC) can certainly contribute to artifacts that may cast into question the veracity of the lactoferrin staining data. To that end, we are including here representative plots for the FSC and SSC of the exact same representative samples used in the neutrophil lactoferrin chart, confirming no significant alteration in either parameter that may falsely influence the lactoferrin staining results.

12. Data is plural.

Yes, certainly. This error has now been corrected.

13. Good literature that for some stimuli the induction of IL-27 is indirect – through type I IFNs but it is less clear how rbc would promote IL-27.

It is an excellent question. The possibilities are multiple, as RBC alone or products of RBC lysis (hemoglobin, heme, or iron), could affect several receptors in microglia (including CD36, CD163, CD91 and iron related pathways). For instance, CD36 (including signaling through MyD88) may lead to NF- κ B activation that could play a role in IL-27 transcription. While exciting, we believe that this question need to be answered independently from the present report. Thanks again for this stimulating comment.

Reviewer #2

1. However, the major criticism of this manuscript is the lack of demonstration whether the protective effect of endogenous and exogenous IL-27 is mediated by LTF. Blockade experiments of LTF in ICH and ICH plus IL-27 would further strengthen the authors' conclusion.

Response here is [REDACTED]

2. To prove that PMNs deliver LTF to the injured brain, the authors used Western blot (Fig. 5e). However, FACS staining of LTF in the brain-infiltrating PMNs as shown in Fig. 5b' or immunohistochemical analysis would be better for it.

We believe that the Western analysis could provide the most direct and highly quantitative assessment of LTF in the ICH-affected brain. This method captures the total brain level of LTF in the tissue, independent of the degree of degranulation that has already occurred and resulted in the depletion of LTF in PMNs. Since we demonstrated that LTF is absent in the normal brain, and that brain cells and other immune cells, except for PMNs are LTF negative (all in agreement with existing literature), we believe that the most probable explanation for the presence of LTF in the brain is its delivery to the ICH-injured brain by PMNs. To strengthen our position, we have now added a panel to fig.5, demonstrating that all of the LTF-positive cells in the ICH-affected brain are neutrophils (Ly6G positive cells).

3. IL-27 consists of p28 and EBI3, and IL-27 receptor consists of IL-27Ra (WSX-1/TCCR) and gp130. Therefore, the authors are required to examine mRNA expression of both molecules in each case (Fig. 1 and 2). In addition, the authors need to clarify whether the ELISA kit can detect heterodimeric IL-27 or p28 alone.

As requested, we have now provided the requested results, which include the characterization of both subunits of IL-27 and both subunits of the receptor, when appropriate. The ELISA kit that we used in our experiment recognizes the IL-27/p28 subunit.

4. In Fig. 2C, intact RBC was used. In contrast, in Fig. 6b, RBC lysate was used. The authors need more explanation on the difference for the readers to easily understand it. Is this because RBC membrane but not lysate is necessary for augmentation of IL-27 expression? If so, the authors may add discussion on the molecular mechanism whereby RBC augments IL-27 expression.

Thanks for pointing out this important issue that we neglected to explain. By using these two models (intact RBC, fig 2c vs. lysed RBC, fig 6), we attempted to mimic the two distinct stages in ICH pathogenesis. Extravasated (consequence of vessel rupture) intact RBCs are the target of phagocytosis/ removal by microglia/macrophages, as a part of the clean-up and repair process. The effective clearance of RBCs may theoretically prevent the hemolysis that normally starts to occur 1-3d after the ICH. However, it is the lysed RBCs (hemolysates, including hemoglobin, heme, and ultimately iron) that possess the highest cytotoxic/neurotoxic potential. In Fig 2C we demonstrated that microglia

produce IL-27 during the engulfment of RBC while in Fig 6b we demonstrated that hemolysates' neurotoxicity is ameliorated by the LTF.

5. *Lane 397, the "immunoblot "should be "immunostaining" or "FACS staining".*

Thank you for the correction; we have now fixed this mistake. FACS staining is now indicated.

6. *Table 1 is missing (lane 235).*

We apologize. Table 1 is now included.

7. *In Fig. 5b', staining with isotype-matched control antibody is necessary.*

We acknowledge the reviewer's inquiry regarding the use of an isotype control, as isotype controls have classically been utilized to determine levels of non-specific binding occurring in an experiment. However, recent literature has moved towards the use of fluorescent-minus-one controls to determine positive and negative gating. <https://www.ncbi.nlm.nih.gov/pubmed/19575390>. As the true purpose of isotype control antibodies is to demonstrate that a sample is correctly block, we have taken every step to reduce non-specific binding in our experiment. We have included both an FC-receptor blocking step and incorporated a 2% proportion of FBS into our flow cytometry media to reduce the degree of non-specific binding common to myeloid cells. We carefully titrated both the primary and secondary antibodies to ensure that positive staining was seen only in neutrophils, with monocytes and T cells serving as negative controls. To that end, there is not ideal way to similarly titrate the isotype control antibody. We are therefore confident that the binding we are seeing for anti-LTF is indeed specific, and have elected to use fluorescence-minus-one controls and the background staining of the secondary antibody (as seen in "miR-1792 family clusters control iNKT cell ontogenesis via modulation of TGF- β signaling (PNAS 2016)) to set the correct positive and negative gates for each of our antibodies." In light of the reviewer's concerns, we have now amended the figure to include the histogram staining from the anti-LTF FMO control (all antibodies, including the fluorescent secondary antibody but lacking anti-LTF) for neutrophils, monocytes and T cells (see Fig.5b'). In addition, in the new flow cytometry experiments performed for our anti-IL27-AB experiments, we have included an IgG isotype control for the a-LTF antibody (see figure 3e), which demonstrates the specificity of the a-LTF antibody. We have also compared the background between including an isotype control antibody and a FMO control, as shown in the figure below, and are confident in the sensitivity of our a-LTF antibody.

Reviewer #3 (CNS path) (Remarks to the Author):

Zhao et al. demonstrate that, in a model of intracerebral hemorrhage (ICH), the immunomodulatory cytokine IL27 may have beneficial effects by promoting iron removal, and by reducing edema and oxidative stress, effects attributed largely to neutrophil-derived lactoferrin. The data lead to two main conclusions: first, neutrophil are not uniformly deleterious in the setting of ICH, second IL27 is a potential therapeutic target for this deadly form of acute stroke.

The paper is of interest since has relevance for both the basic pathology of ICH and to its treatment using approaches to enhance IL27 or its molecular effectors on the brain.

I have the following suggestions for improvement:

1. Abstract: the results on using anti-neutrophil approaches refer to ischemic stroke and not ICH.

Thanks for noticing this inaccuracy! We have now changed the reference for one that is applicable ICH, by Sansing et al.⁶

2. IL27 has also deleterious effects in some settings (Bosmann-Ward, J Leuc Biol, 2013) that should be mentioned upfront.

We agree with the Reviewer that addressing potential therapeutic limitation of IL-27 is important. As such, we have incorporated the following statements: “IL-27 has many activities, including the unique ability to limit inflammation⁷, and immune-mediated pathology associated with autoimmune responses^{22,23}. Elevated expression of IL-27 is reported for some pathological conditions including rheumatoid arthritis⁸, psoriasis⁹, or Multiple Sclerosis¹⁰.”

3. *While neutrophil are a key cell in ICH, other bone marrow derived cells are also the target of IL27, e.g., macrophages (Hirase et al AJP-Heart 2013), and need to be mentioned and/or addressed.*

Thanks for this important comment. Certainly, it is likely that cells other than PMNs can be a target of IL-27 in bone marrow. An example of such cells are macrophages, as indicated in the reference included by the Reviewer. We have now included this important stipulation in our manuscript.

4. *Is IL27 produced in brain (3 hrs) and periphery (spleen 24 hrs) independently? If so what are the cells predominantly involved? The culture data on microglia-neuron-astrocytes are in vitro, which may not reflect the cell-type expression in vivo. The upregulation in spleen is of interest and the cellular localization of the expression would help pinpoint the peripheral IL27 sources. Addressing the relative contribution to the production of IL27 and outcome of ICH by CNS and BM-derived cells would also be of interest, but it may be outside the scope of the paper.*

We have tried to pin-down the identity of cells producing IL-27 using immunofluorescence. Unfortunately, we cannot accept the reliability of staining using this approach. However, to partially address the concerns about the source of IL-27 in the brain, we activated primary neurons, astrocyte, oligodendroglia, and microglia in culture by adding RBC, to produce hemorrhage-like activation of these cells, and observed that only microglia express IL-27. Regarding the spleen, we could certainly do splenectomy to address the relative contribution of the organs. However, it is well-known that spleen plays an instrumental role in pathogenesis of stroke including the release PBCs, platelets, stem cells, and a variety of immune cells. In addition, pre-stroke splenectomy has been demonstrated as protective in context of ischemic and hemorrhagic stroke. Based on this complexity, we believe that measuring the relative contribution of spleen to IL-27 and how this correlates with the ICH outcome could very difficult. As such, we would like to request this Reviewer to accept our paper without the necessity to address this question.

6. *The data on increased IL27 expression with RBC addition in microglia is of interest, but raises the question as to whether RBC would also increase expression in neurons and astrocytes.*

We have now tested this possibility by exposing individual cultured primary brain cells to RBCs. Similar to what we observed in our original submission, microglia were the primary cell-type that demonstrated expression of IL-27 (both p27 and EBi3).

7. *The experiments with neutralizing antibodies are of great interest because they suggest a role of endogenous IL27 in the outcome of ICH. However, one wonders if neutralizing IL27 worsens the outcome of ICH.*

Another excellent comment. We have now performed the experiment with anti-IL-27 antibody *in vivo* and demonstrated that indeed blockade of IL-27 (daily injection of anti-IL27 antibody for 3d) is associated with the worse behavioral outcome. This suggests that IL-27 induction after ICH represents a self-defense mechanism.

8. *Lactoferrin upregulation is a reasonable explanation for the reduction in injury afforded by IL27, but it would be helpful to assess the cellular expression of this protein in brain after ICH.*

Thanks, this is an important issue. Can brain cells generate LTF after ICH? Normally, LTF is synthesized primarily by both the glandular epithelium and neutrophils. However, blood LTF levels are almost exclusively derived from neutrophils¹¹. We believe that LTF is not synthesized in the brain (at least at significant levels), neither normally or after ICH. We have analyzed LTF mRNA in control rat brains and in the brains of animals at various time points after ICH and found no presence of LTF transcription, implying that LTF is not synthesized in the brain - this information has now been included in the manuscript. This strongly suggests that brain LTF (protein) after ICH is due to its delivery to the ICH-affected brain, and is consistent with the fact that LTF protein in neutrophil is primarily synthesized during their maturation in the bone marrow and that LTF renewal by mature neutrophil is limited¹²⁻¹⁷. This may explain why levels of LTF mRNA in mature neutrophils is low and why no increase in LTF mRNA is seen in brain despite neutrophil infiltration and LTF protein deposition. We have now incorporated this information within the discussion section.

9. Is the protective effect of recombinant lactoferrin dose related?

Base on the ongoing studies using a chemically modified lactoferrin molecule (ongoing studies aiming at generating pharmaceutical product), we know that the effect is indeed dose-dependent and that the doses of about 1 to 20 mg/kg show efficacy in ICH in a mouse model of ICH. We are currently verifying these doses using in a porcine model of ICH. We would prefer not to include these data here in order not to detract attention from the central topic (see also answer immediately below), which is to propose that the neutrophils, cells that have unique capacity to synthesize LTF, may generate this substance to benefit ICH-affected brain.

10. The pitfalls of the model of ICH used should be mentioned. Data in another model would enhance the appeal of the paper.

We understand the concerns of the reviewer. There are essentially two experimental models of acute ICH. One model is based on injection of autologous blood to the brain matter and the other uses bacterial collagenase injection to induce intracerebral bleeding. However, collagenase is known to cause profound blood-brain-barrier opening^{18,19} which artificially augments the penetration of the therapeutic agent to the ICH-affected brain tissue, thereby artificially amplifying the access of the therapeutic agent to the brain. This potentially overestimates translational conclusions. Based on our experience, we believe that the blood injection model is the more rigorous and conservative test and as such, is more suitable for translational testing. Despite undeniable differences, primarily associated with the mechanism causing bleeding, there are many similarities between pathophysiology of human ICH and blood injection ICH model. The model simulates: (1) mass effect, (2) clot toxicity, (3) oxidative stress (4) peri-hematoma hypoperfusion, (5) inflammation, (6) edema, (7) inflammatory cells infiltration, (8) white matter damage, and (9) hematoma resolution. Regarding the translational value of our finding, we would like to mention that in addition to the data included in this paper, we have reached database regarding the dose-response, duration of treatment, therapeutic window, sex-independent effect, and the effect in 18-mo old aged mice for LTF-based therapy for ICH. We are currently conducting experiment using a pig ICH model. We also believe that adding this additional data would compromise the focus of this paper, which specifically aims at understanding the mechanisms underlying the role of IL-27, LTF and PMNs in ICH pathophysiology and treatment.

1. Wirtz, S., *et al.* Protection from lethal septic peritonitis by neutralizing the biological function of interleukin 27. *The Journal of experimental medicine* **203**, 1875-1881 (2006).
2. Furusawa, J., *et al.* Promotion of Expansion and Differentiation of Hematopoietic Stem Cells by Interleukin-27 into Myeloid Progenitors to Control Infection in Emergency Myelopoiesis. *PLoS Pathog* **12**, e1005507 (2016).
3. Seita, J., *et al.* Interleukin-27 directly induces differentiation in hematopoietic stem cells. *Blood* **111**, 1903-1912 (2008).
4. Fitzgerald, D.C., *et al.* Suppressive effect of IL-27 on encephalitogenic Th17 cells and the effector phase of experimental autoimmune encephalomyelitis. *Journal of immunology* **179**, 3268-3275 (2007).
5. Liu, J., Guan, X. & Ma, X. Regulation of IL-27 p28 gene expression in macrophages through MyD88- and interferon-gamma-mediated pathways. *The Journal of experimental medicine* **204**, 141-152 (2007).
6. Sansing, L.H., Harris, T.H., Kasner, S.E., Hunter, C.A. & Kariko, K. Neutrophil depletion diminishes monocyte infiltration and improves functional outcome after experimental intracerebral hemorrhage. *Acta neurochirurgica. Supplement* **111**, 173-178 (2011).
7. Bosmann, M. & Ward, P.A. Modulation of inflammation by interleukin-27. *Journal of leukocyte biology* **94**, 1159-1165 (2013).
8. Schmidt, C., *et al.* Expression of interleukin-12-related cytokine transcripts in inflammatory bowel disease: elevated interleukin-23p19 and interleukin-27p28 in Crohn's disease but not in ulcerative colitis. *Inflammatory bowel diseases* **11**, 16-23 (2005).
9. Shibata, S., *et al.* Possible roles of IL-27 in the pathogenesis of psoriasis. *The Journal of investigative dermatology* **130**, 1034-1039 (2010).
10. Senecal, V., *et al.* Production of IL-27 in multiple sclerosis lesions by astrocytes and myeloid cells: Modulation of local immune responses. *Glia* **64**, 553-569 (2016).
11. Hansen, N.E., Malmquist, J. & Thorell, J. Plasma myeloperoxidase and lactoferrin measured by radioimmunoassay: relations to neutrophil kinetics. *Acta medica Scandinavica* **198**, 437-443 (1975).
12. Rado, T.A., Bollekens, J., St Laurent, G., Parker, L. & Benz, E.J., Jr. Lactoferrin biosynthesis during granulocytopenesis. *Blood* **64**, 1103-1109 (1984).
13. Rado, T.A., Wei, X.P. & Benz, E.J., Jr. Isolation of lactoferrin cDNA from a human myeloid library and expression of mRNA during normal and leukemic myelopoiesis. *Blood* **70**, 989-993 (1987).
14. Itoh, K., *et al.* Expression profile of active genes in granulocytes. *Blood* **92**, 1432-1441 (1998).
15. Nagaoka, I., *et al.* Evaluation of the expression of human CAP18 gene during neutrophil maturation in the bone marrow. *Journal of leukocyte biology* **64**, 845-852 (1998).
16. Cowland, J.B. & Borregaard, N. The individual regulation of granule protein mRNA levels during neutrophil maturation explains the heterogeneity of neutrophil granules. *Journal of leukocyte biology* **66**, 989-995 (1999).
17. Mollinedo, F. Human neutrophil granules and exocytosis molecular control. *Immunologia* **22**, 340-358 (2003).
18. Rosenberg, G.A., *et al.* TIMP-2 reduces proteolytic opening of blood-brain barrier by type IV collagenase. *Brain research* **576**, 203-207 (1992).
19. Mun-Bryce, S. & Rosenberg, G.A. Gelatinase B modulates selective opening of the blood-brain barrier during inflammation. *Am J Physiol* **274**, R1203-1211 (1998).

Cover letter addressing the responses to the Reviewer's comments

We thank the reviewers for their outstanding contribution to the equality of this manuscript. The following is the response to their comments:

Reviewer #1 (Remarks to the Author):

Comment: The authors have done a good job of responding to many aspects of the previous critiques. They have performed additional experiments and clarified the manuscript where appropriate. Its a timely study and a neat message about the role of IL-27 in this particular model and does set the stage for more translational studies and I look forward to seeing this develop.

The only very minor point - this is not a gotcha – but can the authors check the isotype of the Biolegend anti-IL-27 Ab? I think it's a mouse IgG2a and they are using a Rat IgG as an isotype control. Probably not a big deal – but this should be clear in the materials and methods.

Response: The Reviewer is correct. We have updated the information in the method section and in Table 2.

Reviewer #2 (Remarks to the Author):

Comment: Since the authors have sufficiently addressed our concerns, I feel that this manuscript is now worthy of publication in the Nature Communication.

Response: We thanks the reviewer for the insightful comments.

Reviewer #3 (Remarks to the Author):

Comment: the authors may want to mention in the discussion the potential mechanisms by which brain injury leads to increases in IL27 in blood and systemic organs. Neurogenic and neurohumoral systems have been described, linking the brain to systemic immunity (J Neurosci. 2015;35:583; Immunol Rev. 2012;248:188) and a brief mention may be of interest to the readers.

Line 67: perhaps "...consistent with the finding...mitigates..." rather than "suggesting"

Line 84: abbreviate IL27 receptor here

Response: As suggested by the Reviewer, we have introduced a short paragraph in the discussion section to elaborate on the possible mechanism by which spleen may experience IL-27 induction in the response to the ICH. Specifically, on page 11 we now say:

“The mechanisms responsible for the increased IL-27 level in the blood and CSF after ICH has not been studied in this project; however, it is possible that IL-27 generated by the ICH-activated microglia could enter the peripheral circulation and or CSF and contribute to its elevated levels. Furthermore, we have observed that after ICH, expression of IL-27 is also elevated in the spleen. While we do not know the routes of communication between ICH-affected brain and spleen, the earlier research demonstrated that chemical (e.g. damage associated molecular pattern molecules, such as HMGB1¹) and/or electrical (efferent neurogenic polysynaptic relay through the vagus nerve²) signals originating from the injured brain could modulate the spleen activities, that could also include IL-27 production”.

Lines 67 and 84; we have incorporated the suggested changes.

1. Liesz, A., *et al.* DAMP signaling is a key pathway inducing immune modulation after brain injury. *The Journal of neuroscience : the official journal of the Society for Neuroscience* **35**, 583-598 (2015).
2. Olofsson, P.S., Rosas-Ballina, M., Levine, Y.A. & Tracey, K.J. Rethinking inflammation: neural circuits in the regulation of immunity. *Immunol Rev* **248**, 188-204 (2012).